# Bias correction of surface downwelling longwave and shortwave radiation for the EWEMBI dataset

Stefan Lange[1]

[1]Potsdam Institute for Climate Impact Research, Telegraphenberg A 31, 14473 Potsdam, Germany

*Correspondence to:* Stefan Lange (slange@pik-potsdam.de)

**Abstract.** Many meteorological forcing datasets include bias-corrected surface downwelling longwave and shortwave radiation (rlds and rsds). Methods used for such bias corrections range from multi-year monthly mean value scaling to quantile mapping at the daily time scale. An additional downscaling is necessary if the data to be corrected have a higher spatial resolution than the observational data used to determine the biases. This was the case when EartH2Observe (E2OBS; Calton et al., 2016)
rlds and rsds were bias-corrected using more coarsely resolved Surface Radiation Budget (SRB; Stackhouse Jr. et al., 2011) data for the production of the meteorological forcing dataset EWEMBI (Lange, 2016). This article systematically compares various parametric quantile mapping methods designed specifically for this purpose, including those used for the production of EWEMBI rlds and rsds. The methods vary in the time scale at which they operate, in their way of accounting for physical upper radiation limits, and in their approach to bridging the spatial resolution gap between E2OBS and SRB. It is shown how
temporal and spatial variability deflation related to bilinear interpolation and other deterministic downscaling approaches can be overcome by downscaling the target statistics of quantile mapping from the SRB to the E2OBS grid such that the sub-SRB-grid scale spatial variability present in the original E2OBS data is retained. Cross-validations at the daily and monthly time scale reveal that it is worthwhile to take empirical estimates of physical upper limits into account when adjusting either radiation component and that, overall, bias correction at the daily time scale is more effective than bias correction at the monthly time
scale if sampling errors are taken into account.

## 1 Introduction

High-quality observational datasets of surface downwelling radiation are of interest in many fields of climate science, including energy budget estimation (Kiehl and Trenberth, 1997; Trenberth et al., 2009; Wild et al., 2013) and climate model evaluation
(Garratt, 1994; Ma et al., 2014; Wild et al., 2015). As part of so-called climate or meteorological forcing datasets such as those generated within the Global Soil Wetness Project (GSWP; Zhao and Dirmeyer, 2003), at Princeton University (Sheffield et al., 2006), and within the WATer and global CHange project (WATCH; Weedon et al., 2011), the longwave and shortwave components of surface downwelling radiation (abbreviated as rlds and rsds or just longwave and shortwave radiation in the

following) are used to, e.g., correct model biases in climate model output (Hempel et al., 2013; Iizumi et al., 2017; Cannon, 2017) and drive simulations of climate impacts (Müller Schmied et al., 2016; Veldkamp et al., 2017; Chang et al., 2017; Krysanova and Hattermann, 2017; Ito et al., 2017).

These meteorological forcing datasets are global, long-term meteorological reanalysis datasets such as those produced by the National Centers for Environmental Prediction-National Center for Atmospheric Research (NCEP-NCAR; Kalnay et al., 1996; Kistler et al., 2001) and the European Centre for Medium-Range Weather Forecasts (ECMWF; Uppala et al., 2005; Dee et al., 2011), refined by bias correction using global, gridded observational data. For the components of surface downwelling radiation, such a bias correction is often necessary because observations of these variables are not assimilated in the reanalyses, which makes them subject to modelling biases of, e.g., land-atmosphere interactions and cloud processes (Kalnay et al., 1996; Ruane et al., 2015).

Different approaches are adopted in order to carry out these bias corrections. Weedon et al. (2011, 2014) apply indirect corrections at the monthly time scale using near-surface air temperature observations for rlds and observations of atmospheric aerosol loadings and cloudiness for rsds. Sheffield et al. (2006) directly rescale rlds and rsds to match observed multi-year monthly mean values. Ruane et al. (2015) directly adjust distributions of daily mean rsds. The observational dataset commonly used for such direct adjustments of rlds and rsds is the Surface Radiation Budget (SRB) dataset assembled by the National Aeronautics and Space Administration (NASA) and the Global Energy and Water EXchanges project (GEWEX; Stackhouse Jr. et al., 2011).

Another meteorological forcing dataset, the EartH2Observe, WFDEI and ERA-Interim data Merged and Bias-corrected for ISIMIP (EWEMBI; Lange, 2016), was recently assembled to be used as the reference dataset for bias correction of global climate model output within the Inter-Sectoral Impact Model Intercomparison Project phase 2b (ISIMIP2b; Frieler et al., 2017). The surface downwelling longwave and shortwave radiation data included in EWEMBI are based on daily rlds and rsds from the climate forcing dataset compiled for the EartH2Observe project (E2OBS; Calton et al., 2016). In order to reduce deviations of E2OBS rlds and rsds statistics from the corresponding SRB estimates in particular over tropical land (Dutra, 2015), for EWEMBI, the former were bias-adjusted to the latter at the daily time scale using two newly developed parametric quantile mapping methods.

These methods are conceptually similar to the Ruane et al. (2015) method, which fits beta distributions to reanalysed and observed daily mean rsds for every calendar month, thereby accounting for upper and lower physical limits of rsds using the multi-year monthly maximum value as the upper and zero as the lower limit of the distribution, and then uses quantile mapping to adjust the distributions. In contrast to Ruane et al. (2015), the methods developed to adjust E2OBS rlds and rsds for EWEMBI applies moving windows to estimate beta distribution parameters for every day of the year. This precludes discontinuities at the turn of the month (Rust et al., 2015; Gennaretti et al., 2015) and promises a better bias correction where the seasonality of radiation is very pronounced such as for rsds at high latitudes. Also, the new methods estimate the physical upper limits of rlds and rsds differently, acknowledging that these limits are necessarily greater than or equal to the greatest value observed during any fixed period. Lastly, while Ruane et al. (2015) linearly interpolate SRB rsds from its natural horizontal resolution of 1.0° to the 0.5° reanalysis grid prior to bias correction, the new methods aggregate the E2OBS data from their original 0.5° grid

to the 1.0° SRB grid, where the bias correction is then carried out, and disaggregates these aggregated and bias-corrected data back to the E2OBS grid. Depending on the disaggregation method, this approach promises to generate bias-corrected data with more realistic temporal as well as spatial variability.

The new methods are comprehensively described and cross-validated in this article. Moreover, several modifications of the new methods are tested here that differ in how they handle the spatial resolution gap between the E2OBS and SRB grids, and how they account for the physical upper limits of rlds and rsds. Also included are bias correction methods that operate at the monthly time scale in order to test if bias correction of daily or monthly mean values yields better overall cross-validation results. The lessons learned from these analyses shall benefit bias corrections of surface downwelling radiation to be carried out in future generations of climate forcing datasets.

## 2  Data

### 2.1  E2OBS

The EartH2Observe (E2OBS; Dutra, 2015; Calton et al., 2016) daily mean rlds and rsds data bias-corrected for EWEMBI cover the whole globe on a regular $0.5° \times 0.5°$ latitude-longitude grid and span the 1979–2014 time period. Over the ocean, E2OBS rlds and rsds are identical to bilinearly interpolated ERA-Interim (ERAI; Dee et al., 2011) rlds and rsds. Over land, they are identical to WATCH Forcing Data methodology applied to ERA-Interim reanalysis data (WFDEI; Weedon et al., 2014) rlds and rsds. WFDEI rlds, in turn, is identical to bilinearly interpolated ERAI rlds, adjusted for elevation differences between the ERAI and Climatic Research Unit (CRU; Harris et al., 2013) grids. WFDEI rsds is identical to bilinearly interpolated ERAI rsds bias-corrected at the monthly time scale using CRU TS3.1/3.21 mean cloud cover and considering effects of interannual changes in atmospheric aerosol optical depths (Weedon et al., 2010, 2011, 2014).

### 2.2  SRB

The observational data used for the bias correction of E2OBS daily mean rlds and rsds for EWEMBI were the NASA-GEWEX Surface Radiation Budget (SRB; Stackhouse Jr. et al., 2011) primary-algorithm estimates of daily mean rlds and rsds from the latest SRB releases available at the time, which were release 3.1 for rlds and release 3.0 for rsds. These data cover the whole globe on a regular $1.0° \times 1.0°$ latitude-longitude grid and span the 07/1983–12/2007 time period. For bias correction and cross-validation, a 24-year subsample of these data was used and is used here that spans the 12/1983–11/2007 time period. Additional data from the adjacent months 11/1983 and 12/2007 are employed for computations of running mean values. The SRB estimates of rlds and rsds are based on satellite-derived cloud parameters and ozone fields, reanalysis meteorology and a few other ancillary datasets. Due to a lack of satellite coverage during most of the 07/1983–06/1998 time period over an area centred at 70°E, SRB data artefacts are present over the Indian Ocean (https://gewex-srb.larc.nasa.gov/common/php/SRB_known_issues.php; cf. Figs. 2–4, 7). Every SRB grid cell contains exactly four E2OBS grid cells.

## 3 Methods

For the reader who is is not familiar with the concepts of quantile mapping and/or statistical downscaling, a short introduction including definitions of relevant terms is given in Appendix A. The parametric quantile mapping methods introduced in the following are named according to the scheme $\mathrm{BC}vtpx$, where $v, t, p$ are used to distinguish between methods for longwave and shortwave radiation ($v = \mathrm{l}, \mathrm{s}$) operating at the daily and monthly time scale ($t = \mathrm{d}, \mathrm{m}$) using basic and advanced distribution types or parameter estimation techniques ($p = \mathrm{b}, \mathrm{a}$). Index $x = 0, 1, 2$ is used for variants of these methods that differ in how they handle the spatial resolution gap between the SRB and E2OBS grids. For the $\mathrm{BC}vtp0$ methods, the SRB data are spatially bilinearly interpolated to the E2OBS grid and the E2OBS data are then bias-corrected using these interpolated SRB data; this is to mimic the Ruane et al. (2015) approach. For bias correction with the $\mathrm{BC}vtp1$ methods, E2OBS data are spatially aggregated to the SRB grid, the aggregated data are then bias-corrected and the resulting data disaggregated back to the E2OBS grid; this approach was used to produce the EWEMBI radiation data. Lastly, the $\mathrm{BC}vtp2$ methods adjust mean values and variances at the E2OBS grid such that mean values and variances of spatial aggregates to the SRB grid match the corresponding SRB estimates while the sub-SRB-grid scale spatial structure of mean values and variances present in the original E2OBS data is retained; this is to overcome the variability deflation induced by the other two approaches. Since the $\mathrm{BC}vtp0$ and $\mathrm{BC}vtp2$ methods are based on the $\mathrm{BC}vtp1$ methods, the latter are introduced first. Readers who are merely interested in how the EWEMBI radiation data were produced are informed that methods $\mathrm{BClda1}$ and $\mathrm{BCsda1}$ were used for that purpose.

### 3.1 Bias correction at the SRB-grid scale

For the $\mathrm{BC}vtp1$ methods, daily mean E2OBS rlds and rsds are first aggregated to the SRB grid using a first-order conservative remapping scheme (Jones, 1999). The conservative remapping ensures that each aggregated value is the grid-cell area-weighted mean of the underlying four E2OBS values. The methods of bias correction of these aggregated values are described in the following. The method used for the subsequent disaggregation to the E2OBS grid is described in Sect. 3.1.3.

The $\mathrm{BC}vtp1$ methods use parametric transfer functions of the form $F_{vtp}^{\mathrm{SRB}-1}(F_{vtp}^{\mathrm{E2OBS}}(\cdot))$, where $F_{vtp}^{\mathrm{E2OBS}}$ and $F_{vtp}^{\mathrm{SRB}}$ are climatological cumulative distribution functions (CDFs) of aggregated E2OBS and SRB data, respectively. The CDFs are estimated individually for every SRB-grid cell and day of the year (Fig. 1). In order to quantify the extent to which bias correction results benefit from explicitly accounting for physical radiation limits, the basic and advanced methods $\mathrm{BCltb1}$ and $\mathrm{BClta1}$ for longwave radiation use normal and beta distributions, respectively. For shortwave radiation, the relevance of physical limits is less questionable, given that the lower limit of zero matters at least during polar night, and that the solar radiation incident upon land and ocean surfaces is limited by the solar radiation incident upon the top of the atmosphere (cf. Fig. 1). Therefore, all $\mathrm{BCstp1}$ methods use beta distributions and the basic and advanced methods only differ in how they estimate the beta distribution parameters (cf. Fig. 1, Table 1).

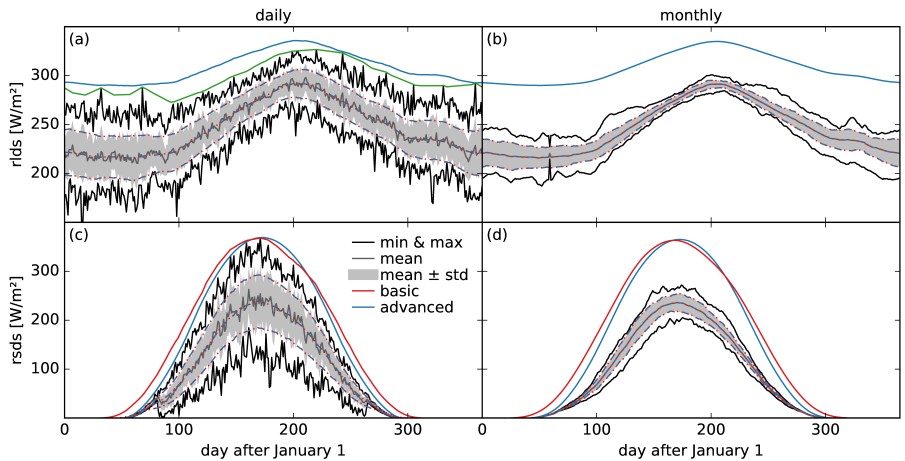

**Figure 1.** Estimation of parameters of quantile mapping methods used for the bias correction of longwave (**top**) and shortwave (**bottom**) radiation at the daily (**left**) and monthly (**right**) time scale. This example is based on SRB daily mean rlds and rsds data from 79.5°N, 12.5°E and the 12/1983–11/2007 time period. Climatological distribution parameters are estimated based on empirical 24-year mean values (dark grey), standard deviations (light grey range around mean values) and minimum and maximum values (black) of daily mean (**left**) and 31-day running mean (**right**) radiation computed for every day of the year. The distribution parameters estimated for the basic (red) and advanced (blue) bias correction methods (cf. Table 1) include mean values and standard deviations (dotted red, dashed blue), and upper bounds (solid red, solid blue) where beta distributions are used. Note that the basic and advanced estimates of mean values and standard deviations only differ in panel (**c**) near the beginning and end of polar night (cf. Table 1). The green line in panel (**a**) represents 25-day running mean values of 25-day running maximum values of 24-year maximum values of daily mean rlds, which are used to estimate the upper bounds of the climatological beta distributions used by the BClda1 method (solid blue line in panel (**a**)). The lower bounds of all climatological beta distributions are set to zero.

### 3.1.1 Bias correction at the daily time scale

The parameters of the climatological CDFs $F_{vdp}^{\mathrm{E2OBS}}$ and $F_{vdp}^{\mathrm{SRB}}$ are estimated based on empirical multi-year mean values, variances and maximum values of daily mean radiation from the 12/1983–11/2007 time period. Data from the whole period were used for the production of EWEMBI rlds and rsds. Data from some half of the period (cf. Sect. 4.1) are used for cross-validation in this study.

For shortwave radiation, the basic daily bias correction method is designed to resemble the method outlined by Ruane et al. (2015, Sect. 3.4). BCsdb1 estimates mean values and variances of climatological beta distributions by 25-day running mean values of multi-year daily mean values and variances, respectively, and their upper bounds by 25-day running mean values of 25-day running maximum values of multi-year maximum values of daily mean rsds (solid red line in Fig. 1c). The idea behind this upper bound estimate is that 25-day running maximum values of multi-year maximum values of daily mean rsds resemble the multi-year monthly maximum values of daily mean rsds used by Ruane et al. (2015). Please note that using the same

**Table 1.** Distribution types and parameter estimation methods of bias correction methods BC$vtp$1 for day $d$ of the year (cf. Fig. 1). Please note that the lower bounds of all climatological beta distributions are set to zero and that 24-year statistics are replaced by 12-year statistics for cross-validation.

| method | distribution type | mean value $\mu_d$ | variance $\sigma_d^2$ | upper bound $b_d$ |
|--------|-------------------|--------------------|-----------------------|-------------------|
| BCldb1 | normal | $\langle\langle x_{ij}\rangle_{i24}\rangle_{j25d}$ | $\langle\{x_{ij}\}_{i24}\rangle_{j25d}$ | — |
| BClda1 | beta | $\langle\langle x_{ij}\rangle_{i24}\rangle_{j25d}$ | $\langle\{x_{ij}\}_{i24}\rangle_{j25d}$ | $A\langle\langle x_{ij}\rangle_{i24}\rangle_{j25d}+B$ |
| BClmb1 | normal | $\langle\langle x_{ij}\rangle_{j31d}\rangle_{i24}$ | $\{\langle x_{ij}\rangle_{j31d}\}_{i24}$ | — |
| BClma1 | beta | $\langle\langle x_{ij}\rangle_{j31d}\rangle_{i24}$ | $\{\langle x_{ij}\rangle_{j31d}\}_{i24}$ | $\langle b_j^{\mathrm{lda1}}\rangle_{j31d}$ |
| BCsdb1 | beta | $\langle\langle x_{ij}\rangle_{i24}\rangle_{j25d}$ | $\langle\{x_{ij}\}_{i24}\rangle_{j25d}$ | $\langle[[x_{ij}]_{i24}]_{j25k}\rangle_{k25d}$ |
| BCsda1 | beta | $\langle\langle x_{ij}\rangle_{i24}\rangle_{j25d*}$ | $\langle\{x_{ij}\}_{i24}\rangle_{j25d*}$ | $C\,\mathrm{rsdt}_d$ |
| BCsmb1 | beta | $\langle\langle x_{ij}\rangle_{j31d}\rangle_{i24}$ | $\{\langle x_{ij}\rangle_{j31d}\}_{i24}$ | $\langle b_j^{\mathrm{sdb1}}\rangle_{j31d}$ |
| BCsma1 | beta | $\langle\langle x_{ij}\rangle_{j31d}\rangle_{i24}$ | $\{\langle x_{ij}\rangle_{j31d}\}_{i24}$ | $\langle b_j^{\mathrm{sda1}}\rangle_{j31d}$ |

$x_{ij}$ is the daily mean rlds (for BCl$tp$1) or rsds (for BCs$tp$1) on day $j$ of year $i$.

Brackets $\langle\cdot\rangle$,$\{\cdot\}$, and $[\cdot]$ denote the calculation of sample mean values, variances, and maximum values, respectively.

Bracket subscripts $i24$, $j31d$, $j25d$, and $j25d^*$ indicate that these sample statistics are calculated over years

$i \in \{1,\ldots,24\}$, over days $j \in \{d-15,\ldots,d+15\}$, over days $j \in \{d-12,\ldots,d+12\}$, and over days

$j \in \{d-n,\ldots,d+n\}$ with $n = \min\{12, \max\{n \geq 0\colon \forall j \in \{d-n,\ldots,d+n\}\colon \mathrm{rsdt}_j > 0\}\}$, respectively.

Constants $A$, $B$, and $C$ are determined by $\arg\min_{A,B'} \sum_{l=1}^{365} (\langle[[x_{ij}]_{i24}]_{j25k}\rangle_{k25l} - A\langle\langle x_{ij}\rangle_{i24}\rangle_{j25l} + B')^2$,

$\min\{B > 0\colon \forall l \in \{1,\ldots,365\}\colon A\langle\langle x_{ij}\rangle_{i24}\rangle_{j25l} + B \geq \langle[[x_{ij}]_{i24}]_{j25k}\rangle_{k25l}\}$, and

$\min\{C > 0\colon \forall j \in \{1,\ldots,365\}\colon C\,\mathrm{rsdt}_j \geq [x_{ij}]_{i24}\}$, respectively.

window length for the running maximum calculation and the additional smoothing ensures that the resulting upper bounds are always greater than or equal to the multi-year maximum values of daily mean rsds.

The BCsda1 method employs the climatology of daily mean shortwave insolation at the top of the atmosphere (rsdt; see Appendix B for how rsdt is calculated in this study) for the upper bound estimation. This is motivated by rsds being limited by
rsdt in most locations and seasons, which suggests that the annual cycle of the upper bound of daily mean rsds has a similar shape as the climatology of daily mean rsdt. Therefore, method BCsda1 uses a rescaled daily mean rsdt climatology as the upper bound climatology of daily mean rsds (solid blue line in Fig. 1c). The rescaling is done with the smallest possible factor that guarantees that the resulting upper bounds are greater than or equal to the multi-year maximum values of daily mean rsds on all days of the year with $\mathrm{rsdt} \geq 50\,\mathrm{W\,m}^{-2}$. An extension of this guarantee to days of the year with lower rsdt would inflate
the rescaling factor because during dusk and dawn of polar night, rsds can exceed rsdt due to diffuse radiation coming in from lower latitudes. Therefore, on days of the year with $\mathrm{rsdt} < 50\,\mathrm{W\,m}^{-2}$, the maximum of the rescaled rsdt and the empirical multi-year maximum daily mean rsds is used as the upper rsds bound. Mean values and variances of the climatological beta distributions of the BCsda1 method are estimated by running mean values of multi-year daily mean values and variances, respectively. The window length used for these running mean calculations is 25 days by default. On days that are fewer than
13 days away from the beginning or end of polar night (as defined by daily mean rsdt going to zero), the window length is shortened to $2n+1$, where $n$ is the number of days between the day in question and the beginning or end of polar night.

For longwave radiation, both the basic and the advanced daily bias correction methods use 25-day running mean values of multi-year daily mean values and variances to estimate climatological mean values and variances, respectively. The upper bounds used by BClda1 are not estimated by the often rather unsmooth 25-day running mean values of 25-day running maximum values of 24-year maximum values of daily mean rlds (solid green line in Fig. 1a) but by a suitably shifted and rescaled mean value climatology (solid blue line in Fig. 1a; formulas in Table 1).

Since the choice of the window length used for all the running mean and maximum value calculations mentioned above is somewhat arbitrary, the window length dependence of the overall performances of the $BCvda1$ methods is investigated in Appendix D. Sensitivities are found to be very low for window lengths between 10 and 40 days.

### 3.1.2 Bias correction at the monthly time scale

In order to mimic a bias correction at the monthly time scale as it was done by, e.g., Sheffield et al. (2006, Sect. 3.d.3), the $BCvmp1$ methods bias-correct 31-day running mean values and then rescale each daily value by the corrected-to-uncorrected ratio of the respective 31-day running mean value.

Mean values and variances of the climatological CDFs $F_{vmp}^{\mathrm{E2OBS}}$ and $F_{vmp}^{\mathrm{SRB}}$ of 31-day running mean values are simply estimated by 24-year (or 12-year for cross-validation) daily mean values and variances of 31-day running mean values, respectively, with February 29 values replaced by averages of February 28 and March 1 values.

Upper bounds of beta distributions are estimated by 31-day running mean values of the upper bounds of the corresponding CDFs $F_{vdp}^{\mathrm{E2OBS}}$ and $F_{vdp}^{\mathrm{SRB}}$ of daily mean radiation (cf. Fig. 1, Table 1) because 31-day running mean values of multi-year maximum values of daily mean radiation are mathematically always greater than or equal to multi-year maximum values of 31-day running mean radiation. The resulting upper bounds are typically much larger than observed 24-year maximum monthly mean radiation (cf. Fig. 1d) because 31 consecutive days of daily mean radiation at the respective physical upper limit are very unlikely to occur in reality.

### 3.1.3 Disaggregation to the E2OBS grid

In principle, the disaggregation of aggregated and bias-corrected E2OBS data from the SRB to the E2OBS grid can be done in various ways. The simplest approach would arguably be a mere interpolation, which is disadvantageous since it ignores the sub-SRB-grid scale spatial variability present in the original E2OBS data. Probabilistic disaggregation methods, on the other hand, that are designed to retain that variability (cf. Sheffield et al., 2006, Sect. 3.b.1), are impractical if, as in the present case, the purpose of the disaggregation is the production and publication of a dataset, because all variants of the dataset that can potentially be generated by a probabilistic algorithm are, as long as all conceivable constraints have been incorporated in the algorithm, equally plausible candidates for the one dataset to be published. Therefore, not a probabilistic but the following deterministic disaggregation approach was used for the production of EWEMBI rlds and rsds and is adopted here for all $BCvtp1$ methods.

First, E2OBS-grid scale upper bounds of daily mean radiation are estimated by bilinearly interpolated maximum values of the climatological upper bounds of SRB all-sky and clear-sky radiation, which in turn are estimated using the BClda1

method for rlds and the $\mathrm{BC}sda1$ methods for rsds (cf. Table 1 and blue lines in Fig. 1a,c). The clear-sky radiation data are included in order to prevent the E2OBS-grid scale upper bounds from being much lower than the real physical limits of daily mean radiation at that spatial scale, given that due to sub-SRB-grid scale spatial variability, upper radiation bounds at the E2OBS-grid scale may exceed those at the SRB-grid scale.

The original daily E2OBS data are then clamped between zero and these upper bounds, and the resulting values (or their distances to their upper bounds) are rescaled day by day and SRB-grid cell by SRB-grid cell such that their SRB-grid scale aggregates match the bias-corrected values. More precisely, for a fixed but arbitrary SRB-grid cell and a fixed but arbitrary day, let $Y$ denote the bias-corrected value at the SRB-grid scale, $w_k$ with $\sum_{k=1}^{4} w_k = 1$ the area weights of the four E2OBS-grid cells $k = 1, 2, 3, 4$ contained in the SRB-grid cell, $X_k$ the clamped original E2OBS data values with upper

bounds $b_k$, and $Y_k$ the bias-corrected values at the E2OBS-grid scale to be computed. If $Y \leq \sum_{k=1}^{4} w_k X_k$, then $Y_k$ is computed according to $Y_k = f X_k$ with $f = Y / \sum_{k=1}^{4} w_k X_k$. Otherwise $Y_k$ is computed according to $Y_k = b_k - f(b_k - X_k)$ with $f = (Y - \sum_{k=1}^{4} w_k b_k) / \sum_{k=1}^{4} w_k (X_k - b_k)$. This rescaling procedure ensures that $0 \leq Y_k \leq b_k$ and $\sum_{k=1}^{4} w_k Y_k = Y$.

### 3.2 Bias correction at the E2OBS-grid scale

#### 3.2.1 The $\mathrm{BC}vtp2$ methods

The disaggregation method introduced above corrects the original E2OBS values from the four E2OBS-grid cells contained in one SRB-grid cell as if they must all be too low (high) if their area-weighted average is too low (high). This implicit assumption is questionable since it rules out the possibility that the area-weighted average is too low because one of the four values is much too low while the others are slightly too high, to give just one example. A statistical manifestation of this problem is illustrated and discussed in Sect. 4.2.

The assumption does not need to be made if the bias correction is carried out directly at the E2OBS grid. With target distributions fixed at the SRB grid, target distributions at the E2OBS grid can be defined such that the bias-corrected data have the SRB-grid scale target distributions and the sub-SRB-grid scale structure of the original E2OBS data. For parametric bias correction methods such as those introduced above, this can be achieved via suitable definitions of the parameters of the E2OBS-grid scale target distributions. Here, for every $\mathrm{BC}vtp1$ method, a corresponding $\mathrm{BC}vtp2$ method is defined to operate

at the same temporal scale and to use the same source (at the E2OBS grid) and target (at the SRB grid) distribution type and parameter estimation technique (cf. Table 1). E2OBS-grid scale target climatologies of mean values, variances and (where necessary) upper bounds are defined as follows.

    The mean value estimates of the original E2OBS data are shifted by a common offset per SRB-grid cell and day of the year to obtain the E2OBS-grid scale target mean values. The offsets are chosen such that the E2OBS-grid scale target mean

values aggregated to the SRB grid match the corresponding SRB mean value estimates. E2OBS data bias-corrected using these E2OBS-grid scale target mean values have SRB grid-scale aggregates that match the SRB grid-scale target mean values because (i) the aggregation is a linear operation and (ii) the mean value of a linear combination of random variables is equal to the same linear combination of the mean values of these random variables.

To obtain the E2OBS-grid scale target variances, the variance estimates of the original E2OBS data are rescaled by a common (to all four E2OBS grid cells contained in one SRB grid cell) factor $f_{ij}$ per day $i$ of the year and SRB-grid cell $j$. For the derivation of the formula for $f_{ij}$ let $Y_{ijk}$ (and $X_{ijk}$) denote random variables representing bias-corrected (and original) E2OBS data from day $i$ of the year and E2OBS-grid cells $k = 1, 2, 3, 4$ contained in SRB-grid cell $j$. Then the estimated variance of the SRB-grid scale aggregate of $Y_{ijk}$ can be expanded to

$$\text{Var}\left(\sum_{k=1}^{4} w_{jk} Y_{ijk}\right) = \sum_{k,l=1}^{4} w_{jk} w_{jl} \text{Cov}(Y_{ijk}, Y_{ijl}) = \sum_{k,l=1}^{4} w_{jk} w_{jl} \text{Cor}(Y_{ijk}, Y_{ijl}) \sqrt{\text{Var}(Y_{ijk}) \text{Var}(Y_{ijl})}, \tag{1}$$

where $w_{jk}$ is the area weight of E2OBS-grid cell $jk$ with $\sum_{k=1}^{4} w_{jk} = 1$ for all $j$, $\text{Cov}(Y_{ijk}, Y_{ijl})$ is the estimated covariance of $Y_{ijk}$ and $Y_{ijl}$, $\text{Cor}(Y_{ijk}, Y_{ijl})$ is the estimated Pearson correlation of $Y_{ijk}$ and $Y_{ijl}$, and $\text{Var}(Y_{ijk})$ is the estimated variance of $Y_{ijk}$. A bias correction would be deemed successful if the left-hand side of Eq. (1) was equal to the estimated variance of $Z_{ij}$, the SRB data from day $i$ of the year and grid cell $j$. On the right-hand side of Eq. (1), $f_{ij} \text{Var}(X_{ijk})$ can be substituted for $\text{Var}(Y_{ijk})$ by definition of the scaling factors, and $\text{Cor}(Y_{ijk}, Y_{ijl})$ can be approximated by $\text{Cor}(X_{ijk}, X_{ijl})$ since quantile mapping preserves ranks and therefore rank correlations and therefore approximately Pearson correlations. The variance scaling factors $f_{ij}$ for method BC$vtp$2 are therefore calculated based on

$$\text{Var} Z_{ij} = f_{ij} \sum_{k,l=1}^{4} w_{jk} w_{jl} \text{Cor}(X_{ijk}, X_{ijl}) \sqrt{\text{Var}(X_{ijk}) \text{Var}(X_{ijl})}, \tag{2}$$

where the variances are estimated using the respective BC$vtp$1 approach (cf. Table 1), and the Pearson correlations are estimated by inversely Fisher-transformed 25-day running mean values of Fisher-transformed 24-year daily Pearson correlations of daily (for BC$vdp$2) or 31-day running mean (for BC$vmp$2) radiation data. The Fisher transformations are invoked here in order to approximately account for correlation value-dependent sampling error intervals (Fisher, 1915, 1921).

The E2OBS-grid scale target upper bounds are calculated in the same way as the E2OBS-grid scale target mean values. This way, the latter rarely exceed the former. Where they do, the latter are reduced to 99 % of the former. For longwave (shortwave) radiation, such reductions are necessary in four (11 % of all) E2OBS grid cells, and there on an average of 15 % (5 %) of all days of the year.

Furthermore, in order to obtain realistic E2OBS-grid scale target beta distributions, the E2OBS-grid scale target variances calculated using Eq. (2) are limited to 40 % of $\mu(b - \mu)$, where $\mu$ and $b$ are the E2OBS-grid scale target mean values and upper bounds, respectively. This limit is imposed because (i) the variance $\sigma^2$ of a random variable taking values from within the interval $[a, b]$ can generally not be greater than $(\mu - a)(b - \mu)$ if $\mu$ is the random variable's mean value, (ii) if that random variable is beta-distributed and $\sigma^2 > (\mu - a)(b - \mu)/2$ then the probability density function is U-shaped (Wilks, 1995), which is considered unrealistic for climatological distributions of rlds and rsds, and (iii) $\sigma^2/(\mu(b - \mu))$ has an empirical upper limit of about 40 % in the original E2OBS radiation data. The 40 % condition is never met for longwave radiation whereas for shortwave radiation it is met in 14 % of all E2OBS grid cells, and there on an average of 2 % of all days of the year.

### 3.2.2 The BC$vtp$0 methods

For the BC$vtp$0 methods, daily SRB data are first bilinearly interpolated to the E2OBS grid. The E2OBS data are then bias-corrected directly at the E2OBS grid using the interpolated SRB data and transfer functions defined exactly as for the respective BC$vtp$1 method.

## 4 Results

In the following, the bias correction methods introduced above are cross-validated at the SRB-grid scale (Sect. 4.1), and their disaggregation performance is assessed by comparing sub-SRB-grid scale spatial variability before and after bias correction (Sect. 4.2).

### 4.1 Cross-validation at the SRB-grid scale

For the cross-validation against SRB data, 24 years worth of overlapping E2OBS and SRB data are divided into two 12-year samples of which the first one is used to calibrate and the second one to validate the method. Common practice would be to use data from the first and second half of the 24-year period to define these samples. Yet due to climate change this definition may yield calibration and validation data samples that differ statistically. These differences in turn, which are essentially climate change signals, may differ in extent between the E2OBS and SRB data. Switanek et al. (2017) have shown that such differences in climate change signals may then dominate cross-validation metrics and thereby distort the comparative validation of bias correction methods. In order to minimise this climate change impact on cross-validation results, here, calibration and validation data samples are composed of data from every second and every other year or vice versa, respectively. The samples are accordingly labelled every1st and every2nd.

Please note that results for BC$vtp$2 are not shown or discussed in this section because BC$vtp$1 and BC$vtp$2 produce virtually identical data at the SRB-grid scale.

### 4.1.1 BC$vtp$0 versus BC$vtp$1

The first question addressed here is how the bilinear spatial interpolation of SRB data to the E2OBS grid before bias correction with the BC$vtp$0 methods impacts the distribution of bias-corrected rlds and rsds values at the SRB-grid scale. To quantify these impacts, biases in multi-year daily mean values, standard deviations, and maximum values remaining after bias correction with methods BC$vda$0 and BC$vda$1 are compared in the left and middle columns of Figs. 2 and 3.

Since linear interpolation always yields values that are intermediate to the values at the interpolation knots it is expected that daily SRB data bilinearly interpolated to the E2OBS grid and then aggregated back up to the SRB grid will be more smooth overall both in space and time than the original SRB data. Manifestations of the increased smoothness in time are the more negative biases of standard deviations (Fig. 2) and maximum values (Fig. 3) remaining after bias correction with BC$vda$0 than with BC$vda$1. Standard deviations after bias correction with BC$vda$0 in particular are negatively biased by more

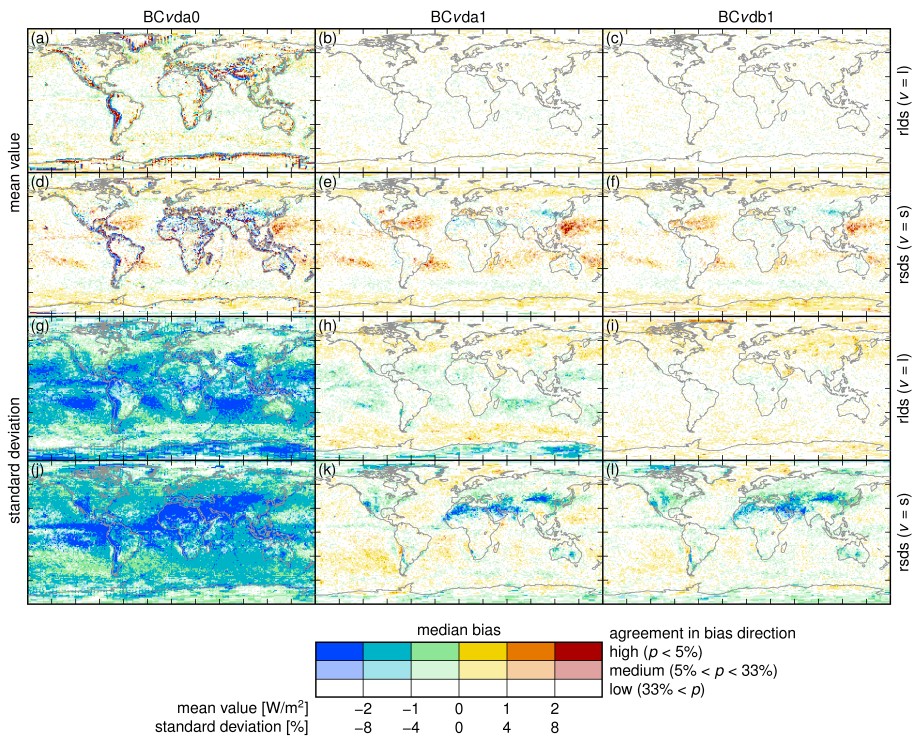

**Figure 2.** Biases relative to SRB in mean values (**a–f**) and standard deviations (**g–l**) of spatially aggregated (to the SRB grid) daily mean longwave (**a–c**, **g–i**) and shortwave (**d–f**, **j–l**) radiation after bias correction with methods BC$v$da0 (**left**), BC$v$da1 (**middle**) and BC$v$db1 (**right**). The biases are calculated individually for each calendar month (January to December) and calibration data sample (every1st, every2nd) pooling SRB and corrected E2OBS data from all years of the corresponding validation data sample (every2nd, every1st, respectively) and omitting shortwave radiation data from months with monthly mean rsdt less than $1\,\mathrm{W\,m^{-2}}$ (cf. Appendix B and Fig. D1c). Depicted are median and agreement in direction (sign of bias) of these individual biases, represented by hue and saturation of a grid cell's colour, respectively. Categories of agreement in bias direction are defined based on one-sided $p$-values obtained from modelling underestimations and overestimations for individual calendar months and validation data samples as outcomes of independent fifty-fifty Bernoulli trials. More saturated colours indicate higher statistical significance of biases remaining after bias correction.

than $4\,\%$ (median over calendar months $\times$ validation data samples) in most regions. In mountainous and therefore spatially heterogeneous regions, also multi-year monthly mean radiation is changed significantly by the interpolation, with median biases over calendar months $\times$ validation data samples remaining after bias correction with BC$v$da0 exceeding $2\,\mathrm{W\,m^{-2}}$ in many such places (Fig. 2).

### 4.1.2 BC$v$ta$x$ versus BC$v$tb$x$

Next is an assessment of how the treatment of the upper bound of the distributions estimated by the BC$v$d$p$1 methods impacts the distribution of bias-corrected rlds and rsds values at the SRB-grid scale. To quantify these impacts, biases in multi-year daily

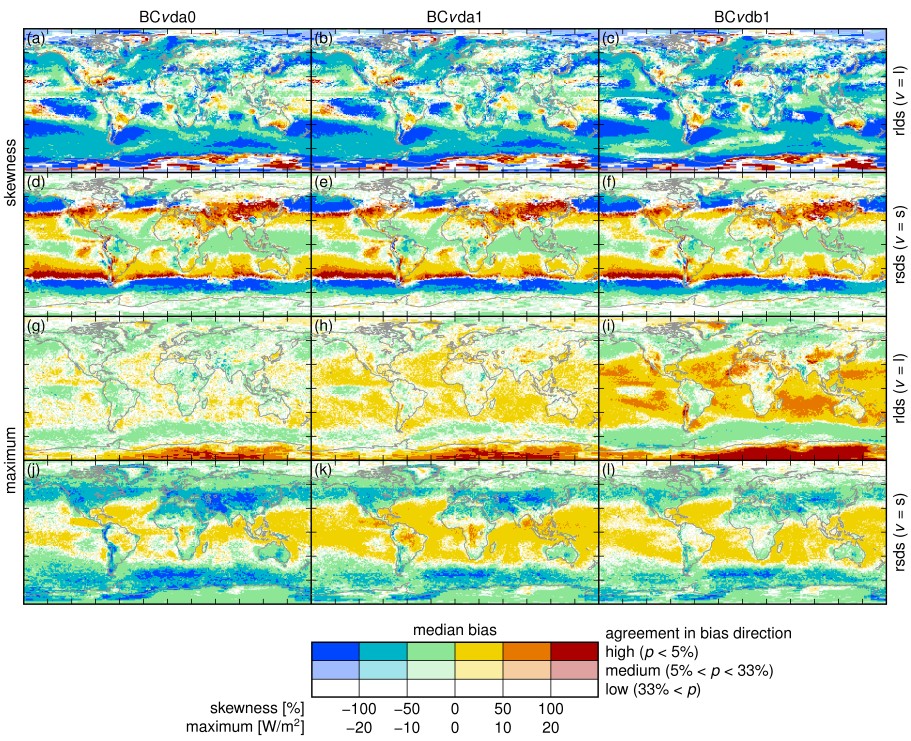

**Figure 3.** Same as Fig. 2 but for biases in skewness (**a–f**) and 12-year maximum values (**g–l**).

mean values, standard deviations, and maximum values remaining after bias correction with methods $BCvda1$ and $BCvdb1$ are compared in the middle and right columns of Figs. 2 and 3.

For longwave radiation, the basic method $BCldb1$ assumes normally distributed values and therefore does not account for any upper physical limit of rlds whereas the advanced method $BClda1$ assumes the existence of such a limit and estimates

5 it empirically. Figure 3 shows that the advanced method generally yields a better correction of 12-year maximum values. In contrast, standard deviations are slightly better corrected by the basic method and mean values are equally well corrected by both methods (Fig. 2).

For shortwave radiation, both the basic and the advanced method empirically estimate upper physical limits of rsds and take these into account in the form of upper bounds of beta distributions. The limit estimates are based on downwelling shortwave

10 radiation at the surface and at the top of the atmosphere for $BCsda1$, and on rsds only for $BCsdb1$. Figure 3 shows that the basic method generally yields a better correction of 12-year maximum values. Also standard deviations and mean values are slightly better corrected by $BCsda1$ than by $BCsdb1$ (Fig. 2).

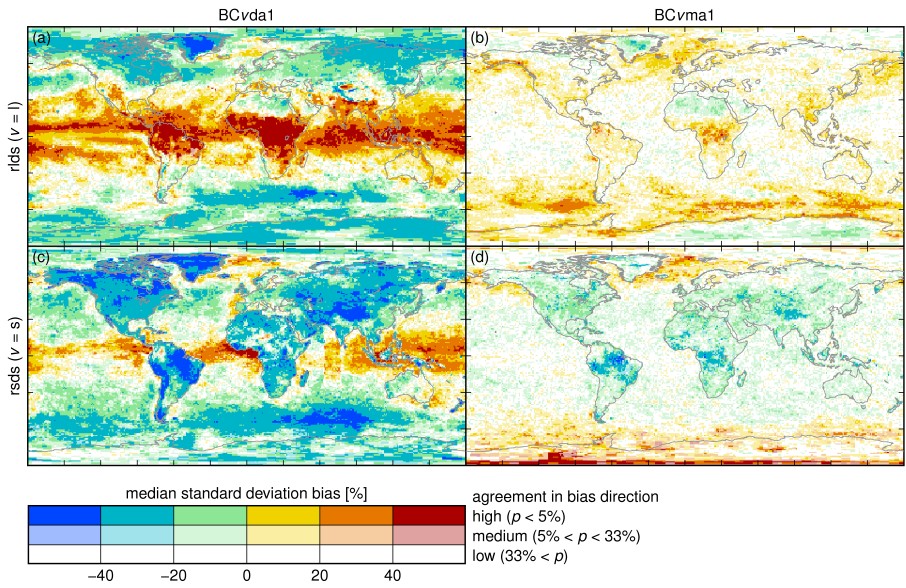

**Figure 4.** Same as Fig. 2 but for relative biases in interannual standard deviations of monthly mean radiation remaining after bias correction with methods BC$v$da1 (**left**) and BC$v$ma1 (**right**).

### 4.1.3  BC$v$d$px$ versus BC$v$m$px$

Next is a comparative cross-validation of methods BC$v$d$px$ and BC$v$m$px$ operating at the daily and monthly time scale, respectively. The cross-validation itself is also done at the daily and monthly time scale based on statistics of daily and monthly mean radiation, respectively. A joint assessment of these cross-validations shall reveal whether bias correction at the daily or

monthly time scale is better overall.

By design, the BC$v$d$px$ and BC$v$m$px$ methods are equally good at correcting multi-year mean values of daily mean radiation. However, both day-to-day and year-to-year variability are expected to be differently well corrected by the methods operating at different time scales. Since day-to-day variability is (not) explicitly adjusted by the methods operating at the daily (monthly) time scale the BC$v$d$px$ methods are expected to perform better at the daily time scale than the BC$v$m$px$ methods.

The year-to-year variability, on the other hand, is explicitly corrected by the BC$v$m$px$ methods and it is not by the BC$v$d$px$ methods because daily data from different years are pooled before quantile mapping is carried out at the daily time scale. Consequently, biases in interannual standard deviations of monthly mean radiation are much larger after bias correction with BC$v$da1 than with BC$v$ma1 (Fig. 4), and the BC$v$m$px$ methods are generally expected to perform better at the monthly time scale than the BC$v$d$px$ methods.

In order to assess whether bias correction at the daily or monthly time scale is more effective overall, a performance measure is needed that is comparable across time scales. Common performance measures of distribution adjustments at individual time scales are the two-sample Kolmogorov-Smirnov (KS) and Kuiper's two-sample test statistic. While Kuiper's test is equally

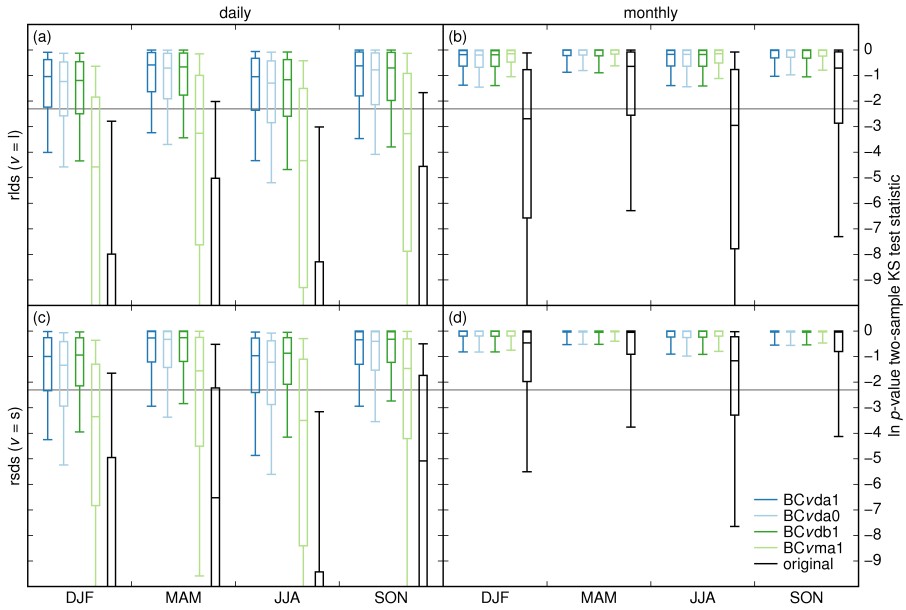

**Figure 5.** Overall performance of bias correction methods $\mathrm{BC}vda1$, $\mathrm{BC}vda0$, $\mathrm{BC}vdb1$, and $\mathrm{BC}vma1$ for longwave (**top**) and shortwave (**bottom**) radiation at the daily (**left**) and monthly (**right**) time scale as quantified by $p$-values of two-sample Kolmogorov-Smirnov test statistics of the respective E2OBS and SRB data before (black) and after (colours) bias correction (cf. Appendix C; greater $p$-values indicate stronger agreement of E2OBS and SRB distributions). The $p$-values are determined individually for each grid cell, season, and calibration data sample, with all corresponding values pooled into one distribution and omitting shortwave radiation data from months with average rsdt less than $1\,\mathrm{W\,m^{-2}}$. The horizontal lines of each box-whisker plot represent the 90th, 75th, 50th, 25th, and 10th (from top to bottom) grid-cell area-weighted percentile of the natural logarithms of these $p$-values over calibration data sample (1sthalf, 2ndhalf), latitude and longitude. The grey horizontal line marks the $p = 10\,\%$ significance level.

sensitive to CDF differences at all quantiles, the KS test is more sensitive at the median than in the tails. A straightforward comparison of these test statistics across time scales is not very meaningful because sample sizes at the daily and monthly time scale differ by a factor of thirty, which implies that the same value of a test statistic has different statistical significance at the daily and monthly time scale. A better comparability can be achieved by comparing the test statistic's $p$-value, which

5  represents the statistical significance of CDF differences. In the present cross-validation, the CDFs compared are based on bias-corrected E2OBS and the corresponding SRB data, and a higher $p$-value indicates more similar CDFs and therefore a better bias correction. For details of the calculation of $p$-values of the two-sample KS and Kuiper's two-sample test statistic see Appendix C.

Global distributions of $p$-values of two-sample test statistics for seasonal distributions of daily and monthly mean rlds and

10  rsds are shown in Fig. 5 for the KS test and Fig. 6 for Kuiper's test. In accordance with expectations, both tests indicate that CDFs are generally better adjusted by $\mathrm{BC}vdpx$ than by $\mathrm{BC}vmpx$ at the daily time scale and vice versa at the monthly time scale. Yet performance differences between $\mathrm{BC}vdpx$ and $\mathrm{BC}vmpx$ are clearly more significant at the daily than at the monthly

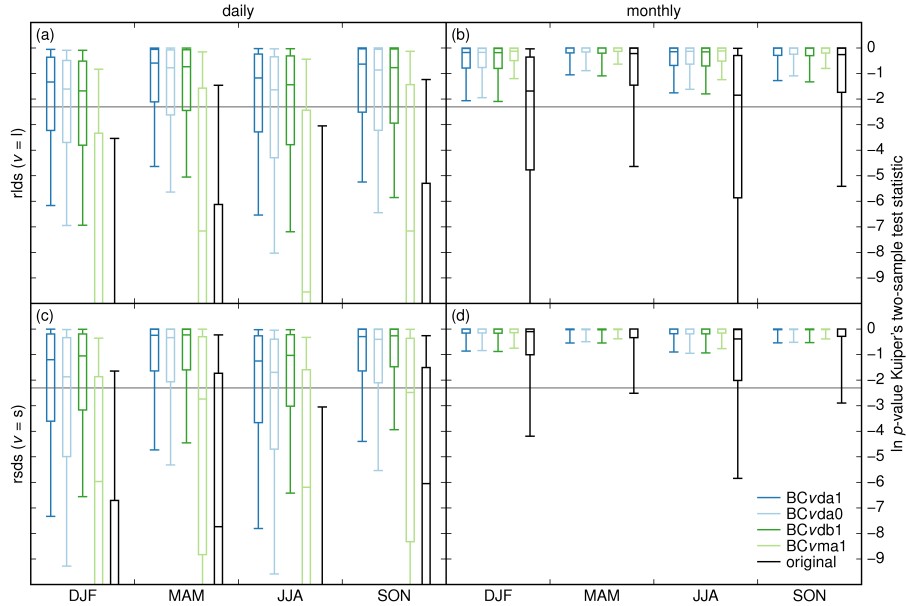

**Figure 6.** Same as Fig. 5 but based on $p$-values of Kuiper's two-sample test statistic.

time scale. This suggests that bias-correcting at the daily instead of at the monthly time scale yields bias decrements at the daily time scale that exceed bias increments at the monthly time scale. Therefore, bias correction at the daily time scale is deemed more effective overall then bias correction at the monthly time scale.

To elaborate this further, the $p = 10\,\%$ significance level is marked by a grey horizontal line in all panels of Figs. 5 and 6 and is to be compared with the 10th percentiles of the global distributions of $p$-values of the two-sample test statistics. Any coincidence of such a 10th percentile with the 10 % significance level suggests that the corresponding $p$-value distribution is in agreement with the null hypothesis of the respective test. Since the null hypothesis of both tests is that the samples compared are from the same underlying distribution, such a coincidence suggests that the bias correction which produced one of the samples compared worked perfectly within the limits of sampling uncertainty. Similarly, 10th percentiles of $p$-value distributions above (below) the 10 % significance level suggest overcorrections (undercorrections) in terms of sampling uncertainty. In that sense, the BC$vtpx$ methods are generally overcorrecting at the monthly time scale and undercorrecting at the daily time scale.

The KS and Kuiper's test statistics also confirm the finding of Sect. 4.1.2 that at the daily time scale, the BC$vda1$ methods outperform the BC$vdb1$ methods for longwave radiation and vice versa for shortwave radiation. This holds true for all seasons and irrespective of CDF differences being generally greater in summer and winter (DJF and JJA) than in the transition seasons (MAM and SON) both before and after bias correction. Moreover, the test statistics find both BC$vda1$ and BC$vdb1$ to outperform BC$vda0$ at the daily time scale, which is in line with the finding of Sect. 4.1.1 that the BC$vda0$ methods deflate day-to-day variability.

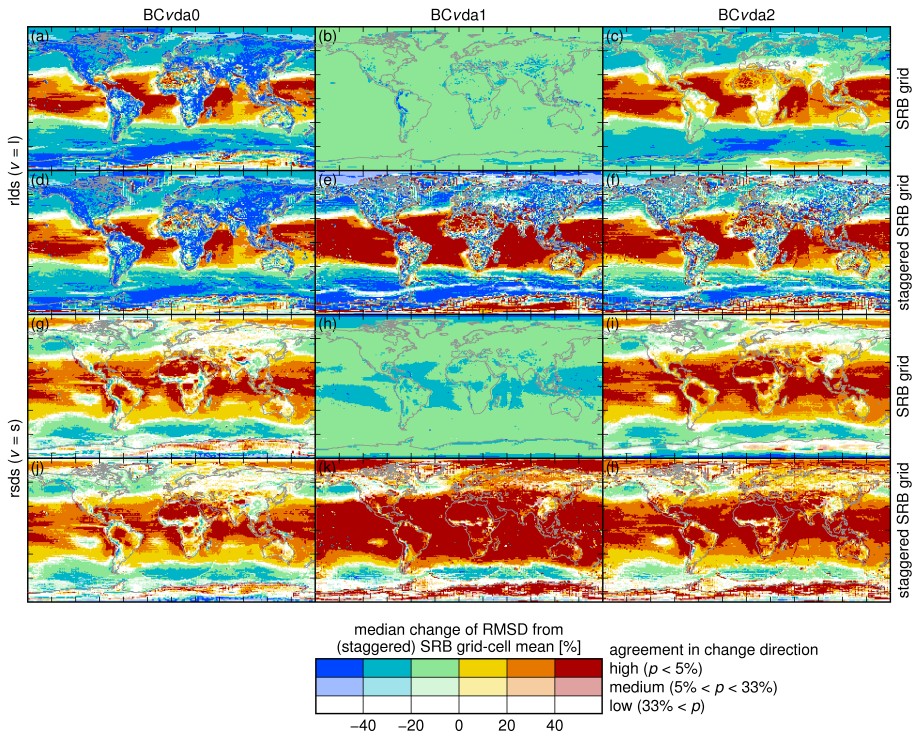

**Figure 7.** Relative change by bias correction with methods $\mathrm{BC}vda0$ (**left**), $\mathrm{BC}vda1$ (**middle**), and $\mathrm{BC}vda2$ (**right**) of the root-mean-square deviation (RMSD) of daily mean E2OBS-grid scale longwave (**a–f**) and shortwave (**g–l**) radiation from the aggregated SRB-grid scale values based on $1°$ grid cells of the SRB grid (**a–c**, **g–i**) and the staggered SRB grid (**d–f**, **j–l**; see text). For every $1°$ grid cell and calendar month, the RMSDs are calculated using original or bias-corrected E2OBS data from the four $0.5°$ grid cells contained in the $1°$ grid cell, pooling data from the entire 12/1983–11/2007 time period and omitting shortwave radiation data from months with average rsdt less than $1\,\mathrm{W\,m}^{-2}$. Depicted are median and agreement in direction of monthly RMSD changes by bias correction (same colouring scheme as in Fig. 2). Very similar results are obtained for the corresponding basic bias correction methods.

The fact that all $\mathrm{BC}vdp1$ methods are undercorrecting at the daily time scale demonstrates the imperfections of these parametric quantile methods. The remaining CDF differences must be linked to imperfect bias corrections of moments of higher than second order since multi-year mean values and standard deviations are well adjusted by design. To illustrate this, relative skewness biases remaining after bias correction with $\mathrm{BC}vdp1$ are shown to exceed 50 % (median over calendar months $\times$ validation data samples) in many regions (Fig. 3). Another manifestation of the imperfections are remaining biases in the tails of the distribution of daily mean rlds and rsds. These must be larger than the remaining median biases because $p$-values of Kuiper's test statistics for these distributions are generally larger than those of the corresponding KS test statistics.

## 4.2 Spatial disaggregation and sub-SRB-grid scale spatial variability

As outlined in Sect. 3.2.1, the $BCvtp1$ approach to the disaggregation of bias-corrected daily mean rlds and rsds values from the SRB- to the E2OBS-grid scale is based on the implicit assumption that the original E2OBS values of daily mean radiation onto the four E2OBS-grid cells contained in one SRB-grid cell must all be too low (high) if their area-weighted average is too low (high). The four original values are then all increased (decreased) by the $BCvtp1$ method. In order to account for their upper (lower) physical bounds, the increases (decreases) are done by a common scaling factor applied to the distances to these bounds. This leads to a reduction of the differences between the four values (necessarily if the four bounds are equal, in most cases if they are similar), i.e., to a deflation of sub-SRB-grid scale spatial variability.

In order to illustrate and quantify the extent of this variability deflation and compare the $BCvtp0$, $BCvtp1$, and $BCvtp2$ methods in terms of their impact on sub-SRB-grid scale spatial variability, the root-mean-square deviation (RMSD) of the four E2OBS-grid scale values of daily mean radiation per SRB-grid cell from their area-weighted average is calculated over all days of a given calendar month both before and after bias correction with either method. Median relative bias correction-induced changes of these RMSDs are depicted in Fig. 7 and demonstrate that $BCvda1$ indeed generally deflates them, in some regions by more than 20 % (median over calendar months) for both longwave and shortwave radiation. In contrast, $BCvtp0$ and $BCvtp2$ deflate or inflate them depending on variable and region.

In an analogous manner, such RMSDs can be computed based on data from the four E2OBS-grid cells contained in one staggered SRB-grid cell, where the staggered SRB grid is a regular $1.0° \times 1.0°$ latitude-longitude grid shifted by $0.5°$ latitude and $0.5°$ longitude relative to the SRB grid, i.e., every staggered SRB-grid cell contains E2OBS-grid cells contained in four different SRB-grid cells. Median relative bias correction-induced changes of these RMSDs are also depicted in Fig. 7. Ideally, bias correction-induced changes of RMSDs from SRB and staggered SRB grid-cell mean values would be equal. It would then be impossible to tell from their comparison whether the bias correction's target distributions were defined on the SRB or on the staggered SRB grid.

The $BCvdp1$ methods do not fulfil this criterion as they deflate RMSDs from SRB-grid cell mean values everywhere while inflating RMSDs from staggered SRB-grid cell mean values in many regions, in particular over the tropical oceans. The criterion is much better fulfilled by the $BCvdp2$ and $BCvdp0$ methods. The RMSDs are generally greater after bias correction with $BCvdp2$ than with $BCvdp0$, i.e., $BCvdp2$ produces data with greater sub-SRB-grid scale spatial variability than $BCvdp0$. This difference is most visible for longwave radiation, for which $BCvdp0$ produces a stark land-sea contrast of RMSD changes with strong RMSD reductions over land whereas $BCvdp0$ does so to a much lesser extent. This strong deflation of sub-SRB-grid scale spatial variability by $BCvtp0$ is believed to be another artefact caused by the bilinear interpolation of SRB data to the E2OBS grid.

## 5 Summary and conclusions

This article introduces various parametric quantile mapping methods for the bias correction of E2OBS daily mean surface downwelling longwave and shortwave radiation using the corresponding SRB data. The quantile mapping methods differ in (i)

the time scale at which they operate, (ii) if and how they take physical upper radiation bounds into account, and (iii) how they handle the spatial resolution gap between E2OBS and SRB.

A cross-validation at the SRB-grid scale demonstrates that statistics of daily mean radiation are better corrected by methods operating at the daily time scale than by methods operating at the monthly time scale, and vice versa for statistics of monthly mean radiation. Since these performance differences are statistically more significant at the daily than at the monthly time scale, overall, bias correction at the daily time scale is deemed more effective then bias correction at the monthly time scale.

The cross-validation further suggests that it is generally worthwhile to explicitly take physical upper radiation bounds into account during quantile mapping. For shortwave radiation, different approaches to their estimation are tested. A simple approach using running maximum values is found to outperform a more complicated one based on daily mean insolation at the top of the atmosphere (rsdt). This must be due to other factors besides rsdt that influence the upper physical bounds of rsds. Atmospheric humidity is an example for such a factor: The highest rsds values usually occur under clear-sky conditions and they are the higher the drier the atmosphere. Atmospheric humidity in turn is limited by the water vapour holding capacity of the atmosphere, which is controlled by atmospheric temperature. The climatology of atmospheric temperature lags that of rsdt. Hence, the climatology of the upper physical bounds of rsds can be expected to deviate from the rsdt climatology.

The cross-validation also reveals to what extent the bilinear spatial interpolation of SRB data to the E2OBS grid prior to bias correction with the $\mathrm{BC}vtp0$ methods deflates day-to-day variability. This variability deflation has a greater effect on bias correction performance than a change of if and how physical upper radiation bounds are taken into account during quantile mapping, but a much smaller effect than a change of the time scale at which the quantile mapping is carried out.

Lastly, the cross-validation at the daily time scale shows that none of the quantile mapping methods tested here is perfect, concerning in particular the adjustment of distribution tails and moments of higher than second order. This indicates that the true distribution of rlds and rsds is not always exactly normal or beta, as assumed by the parametric quantile mapping methods tested here. Potentially, non-parametric quantile mapping methods (that do not rely on such assumptions) could yield better cross-validation results as long as overfitting is avoided (e.g., Gudmundsson et al., 2012). However, an introduction of and comparison to such methods is beyond the scope of this article.

To bridge the spatial resolution gap between E2OBS and SRB, the methods used for the production of EWEMBI rlds and rsds deterministically disaggregate the E2OBS data previously aggregated to and bias-corrected at the SRB grid. It is shown that the method used for that disaggregation introduces artefacts in the sub-SRB-grid scale spatial variability, which can be overcome by applying quantile mapping directly at the E2OBS grid using either bilinearly interpolated SRB data or target distribution parameters that are based on the more coarsely resolved SRB data as well as on sub-SRB-grid scale spatial variability present in the original E2OBS data. This latter approach yields both good cross-validation results at the SRB-grid scale and suitable adjustments of the sub-SRB-grid scale spatial variability.

The best methods identified here are therefore $\mathrm{BClda2}$ for rlds and $\mathrm{BCsdb2}$ for rsds. In comparison to $\mathrm{BClda1}$ and $\mathrm{BCsda1}$ used for the production of EWEMBI rlds and rsds, bias correction with these methods yields more natural sub-SRB-grid scale spatial variability and, in the case of rsds, slightly better cross-validation results at the SRB-grid scale.

*Data availability.* The EWEMBI dataset is publicly available via https://doi.org/10.5880/pik.2016.004.

## Appendix A:  Quantile mapping and statistical downscaling

Quantile mapping is used to adjust the distribution of values from a data sample. In the context of bias correction, the distribution to be adjusted – the source distribution – is believed or known to be more biased than the distribution the source

distribution is adjusted to – the target distribution. In practise, source and target distributions are empirically estimated from the respective samples, in the present case of E2OBS and SRB radiation data, in the form of cumulative distribution functions (CDFs) $F^{\mathrm{E2OBS}}$ and $F^{\mathrm{SRB}}$, respectively. Quantile mapping is then defined by

$$x \mapsto F^{\mathrm{SRB}\,-1}(F^{\mathrm{E2OBS}}(x)), \tag{A1}$$

where $F^{\mathrm{SRB}\,-1}(F^{\mathrm{E2OBS}}(\cdot))$ is called the transfer function.

Quantile mapping is called parametric if the CDFs are assumed to take certain functional forms. Their estimation then reduces to the estimation of the parameters of these functions. Otherwise, quantile mapping is called non-parametric and CDFs are estimated by estimating selected quantiles, between and beyond which quantiles are interpolated and extrapolated, respectively (e.g., Gudmundsson et al., 2012).

In the present study, source and target distributions are assumed to be normal or beta distributions. Mean values and variances

of normal distributions are estimated by running mean values of multi-year daily sample mean values and variances. Lower and upper bounds of beta distributions are set to zero and estimated by physical upper limits of daily mean radiation, respectively. Shape parameters of beta distributions are estimated with the method of moments (Wilks, 1995) using running mean values of multi-year daily sample mean values and variances.

Bias correction includes a spatial disaggregation or downscaling step if the data behind source and target distributions

have different spatial resolution, as in the present case, or represent area mean values and point values, as in the case of quantile mapping between gridded and station data. If the data behind the target distribution have higher resolution/represent finer spatial scales than the data behind the source distribution, then quantile mapping may lead to both temporal and spatial variability inflation (Maraun, 2013). For the reverse case, the present study shows how quantile mapping may lead to both temporal and spatial variability deflation. Maraun (2013) suggests to solve the inflation issue with stochastic downscaling. It

is shown here that the deflation issue of the reverse case can also be overcome with deterministic downscaling at the transfer function level.

## Appendix B:  Daily mean insolation at the top of the atmosphere

Over the course of a year, the total solar irradiance, $S$, varies according to $S = S_0(1 + e\cos(\Theta))^2$, where $S_0 = 1360.8\,\mathrm{W\,m^{-2}}$ is the solar constant (Kopp and Lean, 2011), $e = 0.0167086$ is the Earth's current orbital eccentricity and $\Theta$ is the angle to the

Earth's position from its perihelion, as seen from the Sun. If the orbital angular velocity of the Earth is approximated to vary

sinusoidally in time then the total solar irradiance on day $n$ after January 1 of the first year of a four-year cycle including one leap year is approximately given by

$$S = S_0 \left( 1 + e \cos \left( 2\pi \frac{n-2}{365.25} + 2e \sin \left( 2\pi \frac{n-2}{365.25} \right) \right) \right)^2, \tag{B1}$$

since $S$ is at its maximum when the Earth is at its perihelion, which on average occurs on January 3.

The daily mean insolation at the top of the atmosphere, rsdt, at some fixed geolocation depends on the location's latitude, $\phi$, and on the declination of the Sun, $\delta$, which varies over the course of a year. On day $n$ after January 1 of the first year of a four-year cycle including one leap year, the declination of the sun is approximately given by

$$\sin \delta = \cos \left( 2\pi \frac{n+10}{365.25} + 2e \sin \left( 2\pi \frac{n-2}{365.25} \right) \right) \sin \delta_{\min}, \tag{B2}$$

since $\delta$ is at its minimum value $\delta_{\min} = -23.4392811°$ at the December solstice, which on average occurs on December 22.

Latitude and declination of the Sun determine the hour angle at sunrise, $h$, according to

$$\cos h = \min\{1, \max\{-1, -\tan \phi \tan \delta\}\}. \tag{B3}$$

The daily mean insolation at the top of the atmosphere at latitude $\phi$ on day $n$ is then given by

$$\mathrm{rsdt} = \frac{S}{\pi} (h \sin \phi \sin \delta + \sin h \cos \phi \cos \delta). \tag{B4}$$

For a given latitude, the rsdt climatology used to estimate the upper bounds of the climatological beta distribution of rsds in

the BCsda$x$ methods is derived using Eqs. (B1)–(B4) to compute rsdt over a four-year cycle including one leap year and then averaging calendar day values over the four cases of leap year occurrence in the four-year cycle.

## Appendix C: Two-sample Kolmogorov-Smirnov test and Kuiper's two-sample test

The overall effectivity of the bias correction methods introduced in this study is measured by similarities of empirical CDFs of SRB and E2OBS data before and after bias correction using the two-sample Kolmogorov-Smirnov (KS) test (Kolmogorov,

1933; Smirnov, 1948) and Kuiper's two-sample test (Kuiper, 1962; Stephens, 1965). Let $F_1$ be the empirical CDF of uncorrected or corrected daily or monthly mean longwave or shortwave E2OBS data for one particular grid cell, calendar month and validation data sample, with all corresponding values pooled into one distribution, and let $F_2$ be the empirical CDF of the corresponding SRB data. Then the two-sample KS test statistic, $D$, and Kuiper's two-sample test statistic, $V$, of these CDFs are given by

$$D = \sup_r |F_1(r) - F_2(r)|, \tag{C1}$$

$$V = \sup_r (F_1(r) - F_2(r)) + \sup_r (F_2(r) - F_1(r)). \tag{C2}$$

The null hypothesis of both the KS test and Kuiper's test is that the two data samples whose empirical CDFs are compared have the same underlying distribution. According to Vetterling et al. (1992, Sect. 14.3), the probability $p$ of incorrectly rejecting this null hypothesis can be approximated by

$$p = 1 - F\left(\left[\sqrt{n} + 0.12 + 0.11/\sqrt{n}\right]D\right) \quad \text{and} \tag{C3}$$

$$p = 1 - G\left(\left[\sqrt{n} + 0.155 + 0.24/\sqrt{n}\right]V\right) \tag{C4}$$

for the KS test and Kuiper's test, respectively, where $F$ and $G$ are the CDFs of the asymptotic distributions of $\sqrt{n}D$ and $\sqrt{n}V$, respectively, $n = n_1 n_2/(n_1 + n_2)$ is the effective sample size, and $n_1$ and $n_2$ are the sizes of the samples behind $F_1$ and $F_2$, respectively. This approximation of the true $p$-value is not only asymptotically accurate but already quite good for $n \geq 4$ (cf. Stephens, 1970; Vetterling et al., 1992).

In order to adjust these $p$-values for potential autocorrelations in the samples compared here, which are in fact time series, $n_1$ and $n_2$ in the formula for $n$ are replaced by $n_1(1 - \rho_1)$ and $n_2(1 - \rho_2)$, respectively, as proposed by Xu (2013), where the autoregression coefficients $\rho_1$ and $\rho_2$ of first-order autoregressive processes fitted to the time series are estimated by the respective sample autocorrelation at lag one.

**Appendix D:  Window length for running mean and maximum calculations**

The climatologies of mean values, variances, and upper bounds of daily mean radiation estimated by the $\mathrm{BC}vdpx$ methods are based on running mean values of empirical multi-year daily mean values, variances and running maximum values, respectively. A common window length of 25 days is used for these running mean and maximum value calculations (cf. Table 1). An obvious question is how sensitive the bias correction results are to the choice of this window length.

The question is addressed here via variants of the $\mathrm{BC}vda1$ methods that use uneven window lengths between 10 and 40 days for their running mean and maximum value calculations and are otherwise identical to the $\mathrm{BC}vda1$ method introduced in Sect. 3.1.1. The performance of these $\mathrm{BC}vda1$ variants is then quantified by $p$-values of two-sample KS statistics of bias-corrected E2OBS data cross-validated against SRB data (cf. Sect. 4.1 and Appendix C). The window lengths that maximise these $p$-values vary considerably with location, calendar month and calibration data sample (Fig. D1). The reason for this high variability is illustrated in Fig. D2, where the overall performance of the $\mathrm{BC}vda1$ variants, quantified by $p$-values of two-sample KS statistics aggregated over time (calendar months) and space (grid cells), is shown to only weakly depend on the chosen window length.

The optimal window length is thus highly uncertain. For longwave (shortwave) radiation, the overall performance of the $\mathrm{BC}vda1$ variants is slightly higher for window lengths from the upper (lower) end of the investigated range (Fig. D2). For practical matters, one can apply the methods using any window length between 10 and 40 days and expect similarly well adjusted radiation biases. The choice of 25-day running windows made here for both longwave and shortwave radiation ensures a close-to-optimal performance of the $\mathrm{BC}vda1$ methods for both variables.

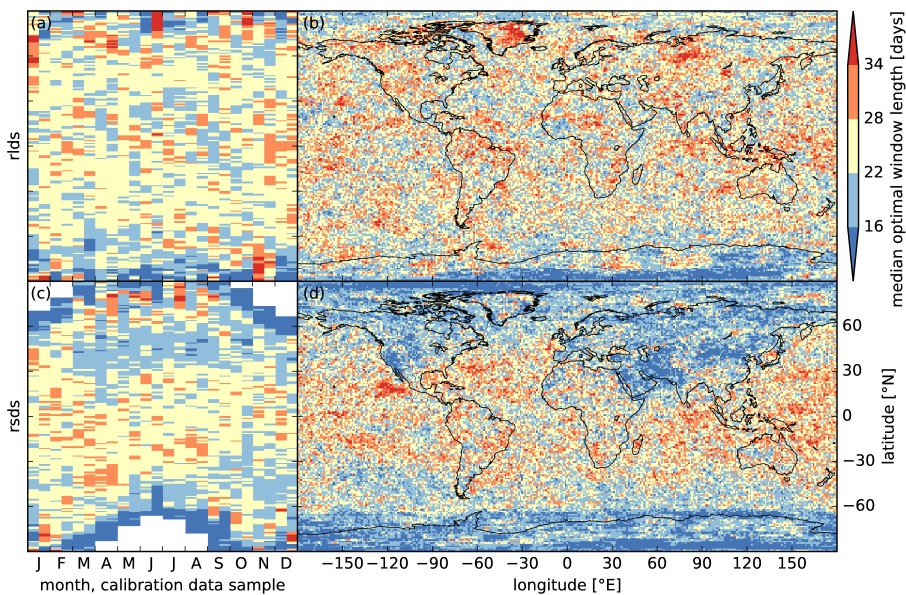

**Figure D1.** Optimal window length for running mean and maximum calculations that precede the estimation of parameters of the climatological distributions of longwave ($v = 1$; **top**) and shortwave ($v = s$; **bottom**) radiation that are used for bias correction with $\mathrm{BC}vda1$ (cf. Table 1). Window lengths are varied between 10 and 40 days. Optimal window lengths maximise the $p$-value of the two-sample KS statistic of bias-corrected E2OBS data cross-validated against SRB data (cf. Sect. 4 and Appendix C) and are determined individually for every grid cell, calendar month (with all corresponding values pooled into one distribution) and calibration data sample (every1st, every2nd). Zonal medians of optimal window lengths for each month and calibration data sample are shown in panels (**a**) and (**c**). Results are masked in (**c**) where and when the monthly mean rsdt (Eqs. (B1)–(B4)) is less than $1\,\mathrm{W\,m^{-2}}$. Panels (**b**) and (**d**) show medians of optimal window lengths over months and calibration data samples.

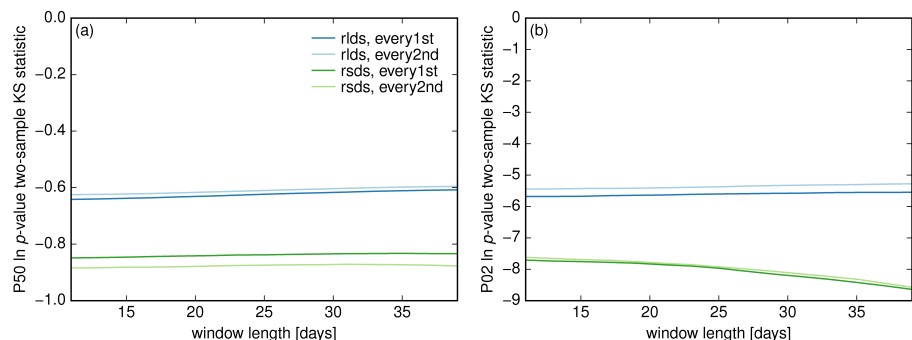

**Figure D2.** Dependence of two-sample KS statistic $p$-values on window length for different radiation types and calibration data samples (see text and Fig. D1). Plotted are the grid-cell area-weighted 50th (**a**) and 2nd (**b**) percentiles of the natural logarithms of the $p$-values over months, latitudes and longitudes.

*Competing interests.*  The author declares that no competing interests are present.

*Acknowledgements.*  The author is grateful to Katja Frieler, Jan Volkholz and Alex Cannon for various helpful discussions at different stages of this work, to Paul W. Stackhouse Jr. for his guidance with SRB data products and the provision of SRB elevation data, to Graham Weedon and Emanuel Dutra for their guidance during the initial stage of assembling the EWEMBI dataset, and to the two anonymous referees who
5   provided highly valuable comments to the discussion paper version of this manuscript. This work has received funding from the European Union's Horizon 2020 research and innovation programme under grant agreement No 641816 Coordinated Research in Earth Systems and Climate: Experiments, kNowledge, Dissemination and Outreach (CRESCENDO).

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
