# Peer review of "Bias correction of surface downwelling longwave and shortwave radiation for the EWEMBI dataset"

_Earth System Dynamics, 2017_

## Referee Comment (RC1) · Anonymous Referee #1 · 1 Nov 2017

The paper compares a large set of methods aimed at correcting, or better saying adjusting, the bias of recently developed EarthH2Observe (E2OBS) shortwave (SW) and longwave (LW) downward surface radiation (rsds and rlds, respectively). These are included in the EWEMBI meteorological forcing dataset for the ISIMIP Intercomparison Project. Bias correction is implemented by comparison with the Surface Radiation Budget (SRB) satellite observational dataset. Given the different spatial resolution of the E2OBS and SRB datasets, an additional downscaling/upscaling is performed in addition to the bias correction itself. The latter is performed by means of quantile mapping, a parametric procedure depending on the parameters that define the distribution adopted for reproducing the statistics of data (i.e. mean values, variance and upper

bound). Disparate methods are thoroughly compared and discussed, varying in the distribution type, time resolution and the way they handle the spatial resolution gap between E2OBS and SRB datasets. An independent validation against station-based observational measurements from the Baseline Surface Radiation Network (BSRN) is also performed.

**1   GENERAL COMMENTS**

Performing bias correction in combination with downscaling/upscaling of model out-puts at observational-based datasets resolution is a well-known critical issue when postprocessing model outputs, and this is well taken into account as background to the analysis. This clearly motivates the careful choice of the variants of the methods, and the thorough discussion of their implications. To me this is a strength and also a limit of this manuscript, since the methodology is constrained by the choice of the model and observational-based assets. Nevertheless such a work provides a necessary reference for further usage of the EWEMBI dataset. For this reason, I would recommend the publication of the manuscript, provided that some minor revisions are provided.

My main concern is that the author provides some improvement to the description of results, particularly in terms of figures. I am aware that comparison among 8*3 methods, adopting different parameters over LW and SW radiation fields separately, requires a challenging effort in terms of clarity and conciseness. In some parts of the manuscript I found difficult to benchmark arguments described in the text with the mentioned figures. I will be more specific in the next section.

Another aspect that I think might be improved is a discussion of the implications of using a deterministic parametric method, rather than a stochastic one, for bias correction when a downscaling/upscaling is made necessary. A reference to Maraun, 2013 (JCLI) might be helpful in this respect. Related to this, a further appendix may be suitable, not only including such a discussion but also a basic description of the quantile mapping

methodology for those who are not familiar with it. In the current draft, this is left to references although, as far as I could check, none of the mentioned papers explicitly addresses for the quantile mapping methodology.

**2  SPECIFIC COMMENTS**

- **Figure 2:** it was very difficult to me to distinguish among the various lines shown in the panels. The dotted red and dashed blue lines are almost undistinguishable (particularly in (b) and (c)) and the light blue line in (a) can hardly be seen. I would suggest to split this figure in two, separately showing the beta and advanced distributions respectively, with the related parameters. As for the caption, I would suggest to explain in first place on which data the computation of the distributions and their parameters is based.

- **Table 2:** I wonder if one could improve the notation for distribution parameters and arrange it with a more mathematically appropriate symbols. Rather than plain text and footnotes, you may want to introduce a consistent notation with brackets and apostrophes to indicate means, running means and variances, as well as apexes and subscripts referring to the length of the window and the amount of years to be considered.

- **l. 32-33, p. 9:** it may be worth mentioning here how the common factor for the aggregation of bias-corrected values in the SRB-grid cell is chosen.

- **l. 22, p. 10:** As far as I understood the common factor f(i,j) is not the same as for the aggregation to the SRB-grid cell, given that it depends on whether the bias correction is applied on the lower or higher resolution. If it is not the case, it is once again not clear to me how the value of this common factor is chosen (see previous comment).

- **l. 33-34 p. 12:** the limits of parametric methods are here correctly mentioned. As stated in the General Comments section, this is a critical issue, and I think it would be worthwhile a few more arguments. If it is not too much work, I wonder if it would be possible to apply a non-parametric quantile mapping (e.g. using a cubic spline empirical CDF) to be compared with these parametric methods.

- **l. 15-16 p. 14:** looking at Figure 6 is very hardly distinguishable that the BCvmp1 at the daily time scale outperforms the same methods at the monthly time scale. This is in my opinion because Figure 6, as well as Figure 2, contains too much information that prevents from emphasizing the key points that are described in the text. The uncertainty range masks the differences among the bars. Furthermore, having five bars for every months makes very difficult to distinguish them, particularly the ones in lighter colours (BCvmp1 methods). I would suggest to split the figures in order at least to separately consider original and bias corrected p-values.

- **l. 9-11 p. 16 and l. 1-2 p. 18:** I found very challenging to carve out the important information from Figures 7-8 and link it with the arguments in the text. It seems to me that the only clear information that can be driven from them is that BCvdax methods outrank BCvdax at the daily resolution for what concerns rlds, and the other way round for what concerns rsds and rlds in the monthly mean. The author refers to a tropical/extratropical asymmetry that to my best effort is barely distinguishable. Furthermore the seasonal dependence (if any) is not mentioned in the text, still making the clarity of the two figures even more arguable. I would suggest either to restructure the layout of Figures 7 and 8 or removing this part, since it does not add much to the discussion of results.

**3 TECHNICAL COMMENTS**

- **l. 6 p. 7 (and elsewhere in the text):** replace "Sect." with "Appendix", when you reference to appendices.

- **l. 5 p. 9:** correct "it".

- **l. 8 p. 10:** maybe "be" is needed between "to" and "made".

- **l. 10 p. 14:** "that" is repeated twice.
* * *

---

## Referee Comment (RC2) · Anonymous Referee #2 · 2 Nov 2017

The author examines different methods for bias correction (BC in the following) of surface downwelling shortwave and longwave radiation (rsds and rlds). More specifically, he uses the Surface Radiation Budget (SRB, 1 degree resolution, 07/1983 - 12/2007) data set to adjust biases in the EartH2Observe (E2OBS, 0.5 degree resolution, 1979 - 2014) data set. The BC methods used differ in the time scale on which they operate (daily or monthly means), the way how the different spatial resolutions (1 degree versus 0.5 degree) are dealt with, and the details of the parametric quantile mapping (beta or normal distribution, use of physical constraints). Two best BC methods, one for longwave the other for shortwave, are identified based on cross-validation and variation considerations. Comparison with independent site-specific measurement data

from the Baseline Surface Radiation Network (BSRN) leads to more mixed results.

The topic of the study - performance of different BC methods for rsds and rlds - is clearly of interest and suitable for ESD. However, in its current form the study suffers from several shortcomings as detailed below. I therefore recommend major revisions.

**General comments:**

A first concern is the focus of the paper: is the focus the evaluation of different methods or the quantitatively correct bias correction rsds and rlds in an absolute sense? Overall, the paper seems to suggest the former (comparison of methods). However, the use of BSRN data as an independent quantitative check points to the later (quantitatively correct rsds and rlds in an absolute sense). If the latter is indeed part of the goal, more work has to go into ascertaining the quantitative correctness of the SRB data used for bias adjustment.

A second major point is the overall clarity of the manuscript. The methods used are complex, the figures shown are (too) packed with interesting information. However, explanations and descriptions come in often (very) long sentences, with lots of details, making it difficult to grasp the essentials. More focused and shorter sentences would help, as would some more information (possibly equations) on the parametric methods. The reason for specific choices (e.g. why comparing these methods, why using these metrics?) are not given. Conclusions read in wide parts more like an extensive summary.

Ideally, the statement that there are two best methods (one for rsds the other for rlds, and measured in terms of cross-validation) would be further embedded. Can these methods be used for bias correction of the entire E2OBS period without introducing artifacts? Could the methods be further improved? Are the other methods just slightly or clearly worse?

**Specific comments:**

**p.3, l.27:** Why use to different versions of SRB for rlds and rsds?

**p.4, l.9:** "If deviations of SRB from SRBQC data quantify methodological uncertainty inherent to SRB data then these findings justify the bias correction of E2OBS rlds and rsds using SRB data over land at least." Two points here. For rsds, one may argue on the same ground that wide parts of the oceans also need adjustment. More generally, you assume here that SRB is correct (at least more correct than E2OBS). How can you be sure? For example, how does SRB compare to CERES data? Or to global mean estimates of rsds and rlds? A number of papers, e.g. by Trenberth et al., give numbers for the latter. An alternative may be to focus only on the methods and not argue at all about the quality of the SRB data.

**Figure 1:** Which of the differences are statistically significant?

**Table 1:** How about the altitude dependence of short wave radiation? (See e.g. Marty, Philipona, Frohlich, Ohmura, Theor. Appl. Climatol. 2002)

**p.6, l.6:** What do you mean by bilinear interpolation from coarse (SRB) to fine (E2OBS) grid? Copying? Same question on p.11, l.18.

**p.6, l.8:** "For the BCvtp2 methods, the sub-SRB-grid scale spatial structure of the original E2OBS data is imposed upon spatially disaggregated SRB data prior to bias correction at the E2OBS grid." Please try to clarify. I think I understood much later, in Section 3.2.1, that you adjust the mean and variance of E2OBS data on the E2OBS grid with mean and variance of SRB data on the corresponding, coarser SRB gird. True?

**p.6, l.14:** "... of the underlying four E2OBS values." The two grids thus are such that four E2OBS cells correspond to one SRB cell? They are not shifted against each other?

**p.6, l.16:** It would be helpful if you added some information, possibly equations, on

[Figure]

transfer functions, target distributions, estimation of means and variances of beta functions etc. in an appendix, as these are absolutely central to your study. Currently, the reader has to know all this or has to check out the references. After all, you even devote an appendix to explaining Kolmogorov-Smirnov.

**Figure 2d:** Why are the colored lines so far away from the black and gray lines?

**p.7, l.8:** What do you mean by "The rsdt climatology at a given latitude is rescaled such that it sits just above the multi-year maximum..."? Why do that?

**p.10, l.9:** "... one possibility to define ..." What would other possibilities be?Why your choice?

**p.10, Eq. 1:** Where does the equation come from? Can you give a reference? The explanation following eq. 1 reads rather lengthy but not too clearly.

**p.11, l.9:** How often does this "99%" condition kick in?

**p.11, l.16:** How often does this "40%" condition kick in?

**p.11, l.27:** "Metrics used..." Why these? Why, for example, skewness? What do I learn from this measure? And why a Kolmogorov-Smirnov test? Why not a test that gives more weight to tails, e.g. Anderson-Darling? More generally, when do you say that your bias adjustment is good? When the adjusted E2OBS distribution is identical (mean, variance, skewness...) to the SRB distribution? Why then adjust at all and not just take the SRB data? Can you use your method to adjust E2OBS data beyond the time span where SRB data is available?

**p.12, l.2:** Does the remark about CVCC imply that your method cannot be used to correct E2OBS data outside the SRD period (1983-2007)?

**p.12, l.11:** "In the following, cross-validation results are only shown and discussed for the BCvtp0 and BCvtp1 methods, since results for the corresponding BCvtp1 and BCvtp2 are virtually identical." What do you mean? That the difference between

BCvtp0 and BCvtp1 is similar as between BCvtp1 and BCvtp2? And, consequently, BCvtp0 and BCvtp2 differ more?

**p.12, l.17:** "... overall performance ..." What do you mean by overall performance?

**p.12, l.24:** Why now looking at relative differences?

**Figure 3:** I guess a good bias correction in your metrics results in a white map. True? The color / hue coding may be better explained upon first use.

**Figures 4 and 5:** Why are the quantities shown of interest? And, again, what is good and what is bad? If white means "good", then none of the methods performs well here?

**p.14, l.15:** Why should bias adjustment on monthly timescales outperform daily bias adjustment with subsequent monthly averaging?

**p.15, l.3:** "Rather, the p-value distributions depicted in Fig. 6b,d suggest that if sampling errors are taken into account then the BCvdp1 methods correct the distributions of monthly mean values almost as well as the BCvmp1 methods." I do not see this point from the text and / or figure.

**p.15, l.7:** "For BCvdp1, this is linked to an insufficient adjustment of third-and higher-order moments..." Not sure what you mean. That you should use another parametric method that takes into account higher moments? At what point do you start to "overfit" if you do this?

**p.15, l.11:** "... correct the upper tail of the rlds and rsds distributions." Can you say this if you use Kolmogorov-Smirnov, which focuses on the center of the distribution?

**Section 4.2:** Comparison with BSRN data. Here you compare point data with area mean data. This comes with potentially quite some uncertainty. See e.g. papers by M.Z. Hakuba et al. 2013 / 2014 / 2016 or N.A.J. Schutgens et al. 2016. Part of your disagreement could have its roots there. More generally, you are looking here more into how good your SRB data is than how good your bias adjustment is. If this is of

interest, you should also consider other data, e.g. CERES or global mean estimates for rlds and rsds, e.g. by Trenberth et al. In its current form, the comparison with BSRN data is rather confusing than helping, I think.

**Figures 7 and 8:** What is the colored rectangle to the lower left in each panel?

**p.18, l.1:** "... and differences between standard deviation biases generated by BCdsdp0, BCsdp1 and BCsdp2 are in line with cross-validation results." What do you mean?

**p.18, l.5:** "... which again suggests that biases relative to BSRN after bias correction using SRB data depend more on the corresponding SRB data biases than on the method used for the bias correction." So the BSRN comparison does not make sense?

**p.18, l.8:** I do not understand this paragraph.

**p.19, l.1 to 14:** I think much of what you are describing here has to do with the fact that you are comparing point measurements with area means. See the above mentioned papers by Hakuba, Schutgens, and references therein.

**p.19, l.26:** Why use a staggered grid?

**Figure 10:** The figure seems to suggest that variability is strongly enhanced (red areas) by the bias adjustment. True?

**Appendix C:** What is the take home message? Figure C2 seems to suggest that the window length is irrelevant. True?
* * *

---

## Author Comment (AC1) · 26 Dec 2017

Responses by the author (in italics) to comments (not in italics) by anonymous referee #1

**General comments**

[...]

My main concern is that the author provides some improvement to the description of results, particularly in terms of figures. I am aware that comparison among 8\*3 methods, adopting different parameters over LW and SW radiation fields separately,

requires a challenging effort in terms of clarity and conciseness. In some parts of the manuscript I found difficult to benchmark arguments described in the text with the mentioned figures. I will be more specific in the next section.

The figures will be clearer in the revised manuscript, see my responses to your specific comments.

Another aspect that I think might be improved is a discussion of the implications of using a deterministic parametric method, rather than a stochastic one, for bias correction when a downscaling/upscaling is made necessary. A reference to Maraun, 2013 (JCLI) might be helpful in this respect. Related to this, a further appendix may be suitable, not only including such a discussion but also a basic description of the quantile mapping methodology for those who are not familiar with it. In the current draft, this is left to references although, as far as I could check, none of the mentioned papers explicitly addresses for the quantile mapping methodology.

I appreciate that not every reader is familiar with the quantile mapping (QM) methodology. Since also anonymous referee #2 asked for it, there will be an appendix in the revised manuscript that shall include a general description of QM and touch on parametric versus non-parametric as well as deterministic versus stochastic QM.

**Specific comments**

Figure 2: it was very difficult to me to distinguish among the various lines shown in the panels. The dotted red and dashed blue lines are almost indistinguishable (particularly in (b) and (c)) and the light blue line in (a) can hardly be seen. I would suggest to split this figure in two, separately showing the beta and advanced distributions respectively, with the related parameters. As for the caption, I would suggest to explain in first place on which data the computation of the distributions and their parameters is based.

As to the caption, I will follow your suggestion. I would not want to split the figure in two as suggested because the figure is supposed to illustrate similarities and differences
between the different QM methods and that would be difficult if different methods were shown in different figures. However, I will simplify the plot by removing the lowermost and uppermost dotted red and dashed blue lines as these are a mere bonus (they just show that the distribution fitting works well). Also, I will make the light blue line green and add the following sentence to the figure caption: "Note that the basic and advanced estimates of mean values and standard deviations only differ in panel (c) near the beginning and end of polar night (cf. Table 1)." This should clarify that it is not a bug but a feature that the dotted red and dashed blue lines are mostly indistinguishable.

Table 2: I wonder if one could improve the notation for distribution parameters and arrange it with a more mathematically appropriate symbols. Rather than plain text and footnotes, you may want to introduce a consistent notation with brackets and apostrophes to indicate means, running means and variances, as well as apexes and subscripts referring to the length of the window and the amount of years to be considered.

Thank you very much for this suggestion. I will introduce such a mathematical notation in the revised manuscript.

I. 32-33, p. 9: it may be worth mentioning here how the common factor for the aggregation of bias-corrected values in the SRB-grid cell is chosen.

I will rewrite this paragraph using a  $g_{ij}$  notation similar to the  $f_{ij}$  notation around equations (1) and (2) such that it becomes clearer how the common factor is determined.

I. 22, p. 10: As far as I understood the common factor f(i,j) is not the same as for the aggregation to the SRB-grid cell, given that it depends on whether the bias correction is applied on the lower or higher resolution. If it is not the case, it is once again not clear to me how the value of this common factor is chosen (see previous comment).

As replied to your previous comment, I will rewrite the paragraph that provoked your previous comment and in that new paragraph use the new notation  $g_{ij}$  for the "common factor" that is indeed different from the "common factor"  $f_{ij}$  mentioned here. After this
adjustment it should be clear how both  $g_{ij}$  and  $f_{ij}$  are calculated.

I. 33-34 p. 12: the limits of parametric methods are here correctly mentioned. As stated in the General Comments section, this is a critical issue, and I think it would be worthwhile a few more arguments. If it is not too much work, I wonder if it would be possible to apply a non-parametric quantile mapping (e.g. using a cubic spline empirical CDF) to be compared with these parametric methods.

The number of QM methods compared in this study is already quite large. Also testing non-parametric QM methods is beyond the scope of the article. However, I will add a paragraph to Section 5 that discusses potential benefits of using non-parametric QM methods compared to the parametric QM methods tested here.

I. 15-16 p. 14: looking at Figure 6 is very hardly distinguishable that the BCvmp1 at the daily time scale outperforms the same methods at the monthly time scale. This is in my opinion because Figure 6, as well as Figure 2, contains too much information that prevents from emphasizing the key points that are described in the text. The uncertainty range masks the differences among the bars. Furthermore, having five bars for every months makes very difficult to distinguish them, particularly the ones in lighter colours (BCvmp1 methods). I would suggest to split the figures in order at least to separately consider original and bias corrected p-values.

Again, I think that it would not help to split the figure as suggested because plotting p-values before and after bias correction using the same scale is needed in order to illustrates the effect of the bias correction. Yet I appreciate that there are quite many box-whisker plots in the figure, so I will reduce the plot's temporal resolution from monthly to seasonal. Also, I will reduce the range of the y-axis from [-14, 0] to [-10, 0], which will make differences between the individual box-whisker plots more easily distinguishable. Lastly, I will add a sentence to the text stating that results for BCvmb1 and BCvma1 are virtually identical.

I. 9-11 p. 16 and I. 1-2 p. 18: I found very challenging to carve out the important
information from Figures 7-8 and link it with the arguments in the text. It seems to me that the only clear information that can be driven from them is that BCvdax methods outrank BCvdax at the daily resolution for what concerns rlds, and the other way round for what concerns rsds and rlds in the monthly mean. The author refers to a tropical/extratropical asymmetry that to my best effort is barely distinguishable. Furthermore the seasonal dependence (if any) is not mentioned in the text, still making the clarity of the two figures even more arguable. I would suggest either to restructure the layout of Figures 7 and 8 or removing this part, since it does not add much to the discussion of results.

Since referee #2 also revealed several substantial shortcomings in this part of the manuscript, the entire validation against BSRN observations will be removed from the revised manuscript.

**Technical corrections**

I. 6 p. 7 (and elsewhere in the text): replace "Sect." with "Appendix", when you reference to appendices.

I will do as suggested.

I. 5 p. 9: correct "it".

I will do as suggested.

I. 8 p. 10: maybe "be" is needed between "to" and "made".

I will do as suggested.

I. 11 p. 14: "that" is repeated twice.

I will substitute "this" for the second "that".

**ESDD**

---

## Author Comment (AC2) · 26 Dec 2017

Responses by the author (in italics) to comments (not in italics) by anonymous referee #2

**General comments**

A first concern is the focus of the paper: is the focus the evaluation of different methods or the quantitatively correct bias correction rsds and rlds in an absolute sense? Overall, the paper seems to suggest the former (comparison of methods). However, the use of BSRN data as an independent quantitative check points to the later (quantitatively

correct rsds and rlds in an absolute sense). If the latter is indeed part of the goal, more work has to go into ascertaining the quantitative correctness of the SRB data used for bias adjustment.

*Many thanks to referee #2 for her comprehensive criticism of the validation agains independent surface observations. After carefully consulting the concerns presented and literature provided by the referee I have decided to completely remove this part of the manuscript. Indeed, the validation was a secondary goal of the paper, which clearly benefits from focusing on its main goal, which is the evaluation of the different quantile mapping (QM) methods.*

A second major point is the overall clarity of the manuscript. The methods used are complex, the figures shown are (too) packed with interesting information. However, explanations and descriptions come in often (very) long sentences, with lots of details, making it difficult to grasp the essentials. More focused and shorter sentences would help, as would some more information (possibly equations) on the parametric methods. The reason for specific choices (e.g. why comparing these methods, why using these metrics?) are not given. Conclusions read in wide parts more like an extensive summary.

*Since referee #1 also pointed to too packed figures, I will reduce their information content to some extent in the revised manuscript and provide more comprehensive explanations, see my responses to your specific comments. I will consult a native English speaker to help shorten sentences where needful. Reasons for choices of methods and metrics will be better motived, see my further responses below. The conclusions section will be distilled to the essentials.*

Ideally, the statement that there are two best methods (one for rsds the other for rlds, and measured in terms of cross-validation) would be further embedded. Can these methods be used for bias correction of the entire E2OBS period without introducing artifacts? Could the methods be further improved? Are the other methods just slightly

or clearly worse?

*The methods can definitely be used for bias correction of the entire E2OBS period, see my response to your specific comment below. Clearly benefitial would be a simultaneous bias correction at both the daily and the monthly time scale. Also, non-parametric QM using upper radiation limits as estimated by my best methods might yield better cross-validation results. These potential improvements will be discussed in the revised manuscript. Qualitative differences between methods will be discussed in more detail.*

**Specific comments**

p.3, l.27: Why use to different versions of SRB for rlds and rsds?

*These are the latest available versions of the SRB dataset. The version numbers differ between rlds and rsds. This will be explained in the revised manuscript.*

p.4, l.9: "If deviations of SRB from SRBQC data quantify methodological uncertainty inherent to SRB data then these findings justify the bias correction of E2OBS rlds and rsds using SRB data over land at least." Two points here. For rsds, one may argue on the same ground that wide parts of the oceans also need adjustment. More generally, you assume here that SRB is correct (at least more correct than E2OBS). How can you be sure? For example, how does SRB compare to CERES data? Or to global mean estimates of rsds and rlds? A number of papers, e.g. by Trenberth et al., give numbers for the latter. An alternative may be to focus only on the methods and not argue at all about the quality of the SRB data.

*In the revised manuscript I will focus only on the methods and not argue at all about the quality of the SRB data.*

Figure 1: Which of the differences are statistically significant?

*This figure will be removed from the manuscript, in line with focusing on the methods.*

Table 1: How about the altitude dependence of short wave radiation? (See e.g. Marty,

[Figure]

Philipona, Frohlich, Ohmura, Theor. Appl. Climatol. 2002)

*Also this table will be removed from the manuscript (and along with it the question of how shortwave radiation changes with altitude), in line with focusing on the methods.*

p.6, l.6: What do you mean by bilinear interpolation from coarse (SRB) to fine (E2OBS) grid? Copying? Same question on p.11, l.18.

*I will change "bilinearly interpolated" to "spatially bilinearly interpolated" in both cases. I think this is a standard term, which does not need further explanation.*

p.6, l.8: "For the BCvtp2 methods, the sub-SRB-grid scale spatial structure of the original E2OBS data is imposed upon spatially disaggregated SRB data prior to bias correction at the E2OBS grid." Please try to clarify. I think I understood much later, in Section 3.2.1, that you adjust the mean and variance of E2OBS data on the E2OBS grid with mean and variance of SRB data on the corresponding, coarser SRB gird. True?

*I will change this sentence to "the BCvtp2 methods adjust mean values and variances at the E2OBS grid such that mean values and variances of spatial aggregates to the SRB grid match the corresponding SRB estimates while the sub-SRB-grid scale spatial structure of mean values and variances present in the original E2OBS data is retained."*

p.6, l.14: "... of the underlying four E2OBS values." The two grids thus are such that four E2OBS cells correspond to one SRB cell? They are not shifted against each other?

*Correct. I will add the sentence "Every SRB grid cell contains exactly four E2OBS grid cells." to the data description section.*

p.6, l.16: It would be helpful if you added some information, possibly equations, on transfer functions, target distributions, estimation of means and variances of beta functions etc. in an appendix, as these are absolutely central to your study. Currently, the reader has to know all this or has to check out the references. After all, you even devote

an appendix to explaining Kolmogorov-Smirnov.

*Thank you for pointing this out. I will add such an appendix to the revised manuscript.*

Figure 2d: Why are the colored lines so far away from the black and gray lines?

*Because my estimates of the upper bounds of monthly mean radiation are calculated based on the upper bounds to the corresponding daily mean radiation. The resulting upper bounds are typically much larger than observed maximum monthly mean radiation because 31 consecutive days of daily mean radiation at its physical upper limit are very unlikely to occur in reality. I will add such an explanation to Sect. 3.1.2.*

p.7, l.8: What do you mean by "The rsdt climatology at a given latitude is rescaled such that it sits just above the multi-year maximum..."? Why do that?

*To answer your questions, I will rewrite the beginning of this paragraph as follows: "The BCsda1 method employs the climatology of daily mean shortwave insolation at the top of the atmosphere (rsdt; see Appendix 1 for how rsdt is calculated in this study) for the upper bound estimation. This is motivated by rsds being limited by rsdt in most locations and seasons, which suggests that the annual cycle of the upper bound of daily mean rsds has a similar shape as the climatology of daily mean rsdt. Therefore, method BCsda1 uses a rescaled daily mean rsdt climatology as the upper bound climatology of daily mean rsds (solid blue line in Fig. 2c). The rescaling is done with the smallest possible factor which guarantees that the resulting upper bounds are greater than or equal to the multi-year maximum values of daily mean rsds on all days of the year with rsdt $\geq$ 50 W m-2. An extension of this guarantee to days of the year with lower rsdt would inflate the rescaling factor because during dusk and dawn of polar night, rsds can exceed rsdt due to diffuse radiation coming in from lower latitudes. On days of the year with rsdt $<$ 50 W m-2, the maximum of the rescaled rsdt and the empirical multi-year maximum daily mean rsds is used as the upper rsds bound."*

p.10, l.9: "... one possibility to define ..." What would other possibilities be? Why your

choice?

*Another possibility would be to follow the BCvtp0 approach, i.e. to use interpolated data. The motivation of my choice is that it solves the problem illustrated and discussed in Sect. 4.3. I will rephrase the sentence as follows: "With target distributions fixed at the SRB grid, target distributions at the E2OBS grid can be defined such that the bias-corrected data have the SRB-grid scale target distributions and the sub-SRB-grid scale structure of the original E2OBS data."*

p.10, Eq. 1: Where does the equation come from? Can you give a reference? The explanation following eq. 1 reads rather lengthy but not too clearly.

*This does not need any reference. It is the standard formula for the variance of a linear combination of random variables. I will however insert one intermediate step using covariances in the equation to make its derivation easier to understand. I will ask a native English speaker to improve the explanation following Eq. 1.*

p.11, l.9: How often does this "99%" condition kick in?

*For longwave (shortwave) radiation, this "99%" condition kicks in over four (about 11% of all) grid cells and there on 15% (about 5%) of all days of the year. I will add this information to the revised manuscript version.*

p.11, l.16: How often does this "40%" condition kick in?

*The "40%" condition is never met for longwave radiation whereas for shortwave radiation it kicks in over 14% of all E2OBS grid cells and there on 2% of all days of the year. I will add this information to the revised manuscript version.*

p.11, l.27: "Metrics used..." Why these? Why, for example, skewness? What do I learn from this measure? And why a Kolmogorov-Smirnov test? Why not a test that gives more weight to tails, e.g. Anderson-Darling? More generally, when do you say that your bias adjustment is good? When the adjusted E2OBS distribution is identical (mean, variance, skewness...) to the SRB distribution? Why then adjust at all and not

just take the SRB data? Can you use your method to adjust E2OBS data beyond the time span where SRB data is available?

*The skewness is included because it is the first distribution moment which is not explicitly adjusted by my parametric QM methods. It is included here to illustrate this conceptual imperfection of my methods. I will include this motivation in the revised manuscript. You are right about the KS test and the relatively low weight it gives to tails. In the revised manuscript version, I will include Kuiper's test as one that (like the suggested AD test) gives the same weight to CDF differences at all quantiles. Qualitatively, however, the Kuiper's test results are the same as those of the KS test. You are right that I (and, as far as I know, everybody else who cross-validates bias correction methods) consider a bias adjustment good if the adjusted distributions are identical to the target distributions. I will include this definition of (overall) performance in the revised manuscript. In the ISIMIP framework, there are two reasons for doing the bias adjustment of E2OBS to SRB data and not just using the SRB data directly: It (i) promises a higher inter-variable consistency (e.g. consistency of temperature and longwave radiation) in the EWEMBI dataset and (ii) prodces radiation data that cover a longer time span. Applying the methods to E2OBS data beyond the time span where SRB data are available is fine since the 1979–2013 period is in fact not much larger than the 1983–2007 period, so that the former is expected to be sufficiently well represented by the latter.*

p.12, l.2: Does the remark about CVCC imply that your method cannot be used to correct E2OBS data outside the SRB period (1983–2007)?

*No, it does not, see my response to your previous comment.*

p.12, l.11: "In the following, cross-validation results are only shown and discussed for the BCvtp0 and BCvtp1 methods, since results for the corresponding BCvtp1 and BCvtp2 are virtually identical." What do you mean? That the difference between BCvtp0 and BCvtp1 is similar as between BCvtp1 and BCvtp2? And, consequently, BCvtp0 and BCvtp2 differ more?

*No, I mean that cross-validation results for the BCvtp1 and BCvtp2 methods are virtu-ally identical. I will remove the word "corresponding" as I suspect that the confusion will be gone once this word is gone. I will also consult a native English speaker about this.*

p.12, l.17: "... overall performance ..." What do you mean by overall performance?

*This will be answered in the introduction part of the results section of the revised manuscript, see my response to your comment on p.11, l.27.*

p.12, l.24: Why now looking at relative differences?

*Why not? For standard deviations, I think this makes more sense than to look at absolute differences.*

Figure 3: I guess a good bias correction in your metrics results in a white map. True? The color / hue coding may be better explained upon first use.

*Not true. White means low agreement in bias direction (positive or negative bias) over months and validation data samples. I will consult a native English speaker to see how the explanation can be improved.*

Figures 4 and 5: Why are the quantities shown of interest? And, again, what is good and what is bad? If white means "good", then none of the methods performs well here?

*The cross-validation of multi-year maximum values shall reveal if it is worthwhile and if so, then how to explicitly adjust upper radiation bounds. I will include this sentence in the introduction part of the results section of the revised manuscript. For why skewness is of interest, see my response to your comment on p.11, l.27. As to the significance of Figure 5, see my answer to your next question. In terms of what white means, see my answer to your previous comment. The methods are clearly not perfect but I also did not expect that. It does not make sense to make an absolute statement such as "this shows that the method performs well." The only sensible question is if one method performs better than another. Figures 4 and 5 quantify the magnitude of biases of*

*selected statistics that remain after bias correction with different methods.*

p.14, l.15: Why should bias adjustment on monthly timescales outperform daily bias adjustment with subsequent monthly averaging?

*Because of what is shown in Figure 5. I will revise the explanation of Figure 5 earlier in the text as follows in order to answer your question: "At the monthly time scale, lower biases are expected to remain after bias correction at the monthly than at the daily time scale. Most importantly, the interannual variability of monthly mean radiation is explicitly adjusted by the BCvmpx methods whereas it is not by the BCvdpx methods as the latter pool daily mean radiation values from all years before the adjustment and are therefore oblivious to variability at the interannual time scale. As an example, in Fig. 5, median biases of interannual standard deviations of monthly mean rlds and rsds are shown to be mostly within/beyond +-20% after bias correction with BCvma1/BCvda1."*

p.15, l.3: "Rather, the p-value distributions depicted in Fig. 6b,d suggest that if sampling errors are taken into account then the BCvdp1 methods correct the distributions of monthly mean values almost as well as the BCvmp1 methods." I do not see this point from the text and / or figure.

*I will remove this sentence from the revised manuscript.*

p.15, l.7: "For BCvdp1, this is linked to an insufficient adjustment of third-and higher-order moments..." Not sure what you mean. That you should use another parametric method that takes into account higher moments? At what point do you start to "overfit" if you do this?

*I mean that my parametric methods explicitly adjust mean values and variances. Higher-order moments are only implicitly (and therefore most likely not perfectly) adjusted through the distribution fitting. In fact, with my methods you cannot overfit in your sense because both the normal and the beta (provided its bounds have been fixed) distribution only have two parameters, which are fixed once two moments have*

[Figure]

*been fixed. Therefore, they cannot adjust more than two moments explicitly. An alter-native would be to use non-parametric QM methods. I think that all of this will become clearer thanks to the appendix about QM and downscaling that will be appended to the revised manuscript.*

p.15, l.11: "... correct the upper tail of the rlds and rsds distributions." Can you say this if you use Kolmogorov-Smirnov, which focuses on the center of the distribution?

*Kuiper's test gives the same result. I will adjust the statement accordingly.*

Section 4.2: Comparison with BSRN data. Here you compare point data with area mean data. This comes with potentially quite some uncertainty. See e.g. papers by M.Z. Hakuba et al. 2013 / 2014 / 2016 or N.A.J. Schutgens et al. 2016. Part of your disagreement could have its roots there. More generally, you are looking here more into how good your SRB data is than how good your bias adjustment is. If this is of interest, you should also consider other data, e.g. CERES or global mean estimates for rlds and rsds, e.g. by Trenberth et al. In its current form, the comparison with BSRN data is rather confusing than helping, I think.

*I agree (see above). I will remove this section from the manuscript.*

Figures 7 and 8: What is the colored rectangle to the lower left in each panel?

*These figures will be removed from the revised manuscript.*

p.18, l.1: "... and differences between standard deviation biases generated by BCdsdp0, BCsdp1 and BCsdp2 are in line with cross-validation results." What do you mean?

*Irrelevant now that this part will be removed from the revised manuscript.*

p.18, l.5: "... which again suggests that biases relative to BSRN after bias correction using SRB data depend more on the corresponding SRB data biases than on the method used for the bias correction." So the BSRN comparison does not make sense?

*Irrelevant now that this part will be removed from the revised manuscript.*

p.18, l.8: I do not understand this paragraph.

*Irrelevant now that this part will be removed from the revised manuscript.*

p.19, l.1 to 14: I think much of what you are describing here has to do with the fact that you are comparing point measurements with area means. See the above mentioned papers by Hakuba, Schutgens, and references therein.

*Maybe. Will be removed from the revised manuscript.*

p.19, l.26: Why use a staggered grid?

*Smaller differences between RMSDs of adjusted E2OBS data from SRB-grid cell and staggered SRB-grid cell mean values are considered to indicate a better bridging of the E2OBS-to-SRB spatial scale gap. Ideally, there would be no such difference and it would therefore be impossible to tell from this analysis if the target distributions of the bias correction were defined on the SRB or staggered SRB grid. I will include this explanation in the revised manuscript.*

Figure 10: The figure seems to suggest that variability is strongly enhanced (red areas) by the bias adjustment. True?

*True.*

Appendix C: What is the take home message? Figure C2 seems to suggest that the window length is irrelevant. True?

*True.*

---

## Author Response (AR1)

**Responses by the author (in green) to comments (in black) by anonymous referee #1**

1. General Comments

**[...]**

My main concern is that the author provides some improvement to the description of results, particularly in terms of figures. I am aware that comparison among 8\*3 methods, adopting different parameters over LW and SW radiation fields separately, requires a challenging effort in terms of clarity and conciseness. In some parts of the manuscript I found difficult to benchmark arguments described in the text with the mentioned figures. I will be more specific in the next section.

The results section has been almost completely rewritten and the figures have been made clearer, see my responses to your specific comments.

Another aspect that I think might be improved is a discussion of the implications of using a deterministic parametric method, rather than a stochastic one, for bias correction when a downscaling/upscaling is made necessary. A reference to Maraun, 2013 (JCLI) might be helpful in this respect. Related to this, a further appendix may be suitable, not only including such a discussion but also a basic description of the quantile mapping methodology for those who are not familiar with it. In the current draft, this is left to references although, as far as I could check, none of the mentioned papers explicitly addresses for the quantile mapping methodology.

I appreciate that not every reader is familiar with the quantile mapping (QM) methodology. Since also anonymous referee #2 asked for it, I have added Appendix A that includes a general description of QM and touches on parametric versus non-parametric as well as deterministic versus stochastic QM.

**2. Specific Comments**

Figure 2: it was very difficult to me to distinguish among the various lines shown in the panels. The dotted red and dashed blue lines are almost indistinguishable (particularly in (b) and (c)) and the light blue line in (a) can hardly be seen. I would suggest to split this figure in two, separately showing the beta and advanced distributions respectively, with the related parameters. As for the caption, I would suggest to explain in first place on which data the computation of the distributions and their parameters is based.

As to the caption, I followed your suggestion. I did not want to split the figure in two as suggested because the figure is supposed to illustrate similarities and differences between the different QM methods and that would be difficult if different methods were shown in different figures. However, I have simplified the plot by removing the lowermost and uppermost dotted red and dashed blue lines as these were a mere bonus (they just showed that the distribution fitting works well). Also, I have made the light blue line green and added the following sentence to the figure caption: "Note that the basic and advanced estimates of mean values and standard deviations only differ in panel (c) near the beginning and end of polar night (cf. Table 1)." This should clarify that it is not a bug but a feature that the dotted red and dashed blue lines are mostly indistinguishable.

Table 2: I wonder if one could improve the notation for distribution parameters and arrange it with a more mathematically appropriate symbols. Rather than plain text and footnotes, you may want to introduce a consistent notation with brackets and apostrophes to indicate means, running means and variances, as well as apexes and subscripts referring to the length of the window and the amount of

years to be considered.

Thank you very much for this suggestion. I have introduced such a mathematical notation in the revised manuscript.

l. 32-33, p. 9: it may be worth mentioning here how the common factor for the aggregation of biascorrected values in the SRB-grid cell is chosen.

**I have rewritten this paragraph such that it is now clearer how the common factor is determined.**

l. 22, p. 10: As far as I understood the common factor f(i,j) is not the same as for the aggregation to the SRB-grid cell, given that it depends on whether the bias correction is applied on the lower or higher resolution. If it is not the case, it is once again not clear to me how the value of this common factor is chosen (see previous comment).

**I have also adjusted this part such that it should be clear how $f_{ij}$ is calculated.**

l. 33-34 p. 12: the limits of parametric methods are here correctly mentioned. As stated in the General Comments section, this is a critical issue, and I think it would be worthwhile a few more arguments. If it is not too much work, I wonder if it would be possible to apply a non-parametric quantile mapping (e.g. using a cubic spline empirical CDF) to be compared with these parametric methods.

The number of QM methods compared in this study is already quite large. Also testing nonparametric QM methods is beyond the scope of the article. However, I have added a paragraph to Section 5 that discusses potential benefits of using non-parametric QM methods compared to the parametric QM methods tested here.

l. 15-16 p. 14: looking at Figure 6 is very hardly distinguishable that the BCvmp1 at the daily time scale outperforms the same methods at the monthly time scale. This is in my opinion because Figure 6, as well as Figure 2, contains too much information that prevents from emphasizing the key points that are described in the text. The uncertainty range masks the differences among the bars. Furthermore, having five bars for every months makes very difficult to distinguish them, particularly the ones in lighter colours (BCvmp1 methods). I would suggest to split the figures in order at least to separately consider original and bias corrected p-values.

Again, I think that it would not help to split the figure as suggested because plotting p-values before and after bias correction using the same scale is needed in order to illustrates the effect of the bias correction. Yet I appreciate that there are quite many box-whisker plots in the figure, so I have reduced the plot's temporal resolution from monthly to seasonal. Also, I have reduced the range of the y-axis from [-14, 0] to [-10, 0], which has made differences between the individual box-whisker plots more easily distinguishable.

l. 9-11 p. 16 and l. 1-2 p. 18: I found very challenging to carve out the important information from Figures 7-8 and link it with the arguments in the text. It seems to me that the only clear information that can be driven from them is that BCvdax methods outrank BCvdax at the daily resolution for what concerns rlds, and the other way round for what concerns rsds and rlds in the monthly mean. The author refers to a tropical/extratropical asymmetry that to my best effort is barely distinguishable. Furthermore the seasonal dependence (if any) is not mentioned in the text, still making the clarity of the two figures even more arguable. I would suggest either to restructure the layout of Figures 7 and 8 or removing this part, since it does not add much to the discussion of results.

Since referee #2 also revealed several substantial shortcomings in this part of the manuscript, the entire validation against BSRN observations has been removed from the revised manuscript.

3. Technical comments

l. 6 p. 7 (and elsewhere in the text): replace "Sect." with "Appendix", when you reference to appendices.

I have done as suggested.

l. 5 p. 9: correct "it".

I have done as suggested.

l. 8 p. 10: maybe "be" is needed between "to" and "made".

I have done as suggested.

l. 11 p. 14: "that" is repeated twice.

I have substituted "this" for the second "that".

**Responses by the author (in green) to comments (in black) by anonymous referee #2**

**1. General comments**

A first concern is the focus of the paper: is the focus the evaluation of different methods or the quantitatively correct bias correction rsds and rlds in an absolute sense? Overall, the paper seems to suggest the former (comparison of methods). However, the use of BSRN data as an independent quantitative check points to the later (quantitatively correct rsds and rlds in an absolute sense). If the latter is indeed part of the goal, more work has to go into ascertaining the quantitative correctness of the SRB data used for bias adjustment.

Many thanks to referee #2 for her comprehensive criticism of the validation agains independent surface observations. After carefully consulting the concerns presented and literature provided by the referee I have decided to completely remove this part of the manuscript. Indeed, the validation was a secondary goal of the paper, which clearly benefits from focusing on its main goal, which is the evaluation of the different quantile mapping (QM) methods.

A second major point is the overall clarity of the manuscript. The methods used are complex, the figures shown are (too) packed with interesting information. However, explanations and descriptions come in often (very) long sentences, with lots of details, making it difficult to grasp the essentials. More focused and shorter sentences would help, as would some more information (possibly equations) on the parametric methods. The reason for specific choices (e.g. why comparing these methods, why using these metrics?) are not given. Conclusions read in wide parts more like an extensive summary.

Since referee #1 also pointed to too packed figures, I have reduced their information content to some extent in the revised manuscript. Also, I have almost entirely rewritten the results and conclusions sections using shorter sentences. These parts are now better structured, more focused and concise. Reasons for choices of methods and metrics are now better motivated.

Ideally, the statement that there are two best methods (one for rsds the other for rlds, and measured in terms of cross-validation) would be further embedded. Can these methods be used for bias correction of the entire E2OBS period without introducing artifacts? Could the methods be further improved? Are the other methods just slightly or clearly worse?

The methods can definitely be used for bias correction of the entire E2OBS period, see my response to your specific comment below. The relative performances of the different methods are now better described in the conclusions section.

**2. Specific comments**

p.3, l.27: Why use to different versions of SRB for rlds and rsds?

These are the latest available versions of the SRB dataset. The version numbers differ between rlds and rsds. This is now explained.

p.4, l.9: "If deviations of SRB from SRBQC data quantify methodological uncertainty inherent to SRB data then these findings justify the bias correction of E2OBS rlds and rsds using SRB data over land at least." Two points here. For rsds, one may argue on the same ground that wide parts of the oceans also need adjustment. More generally, you assume here that SRB is correct (at least more correct than E2OBS). How can you be sure? For example, how does SRB compare to CERES data?

Or to global mean estimates of rsds and rlds? A number of papers, e.g. by Trenberth et al., give numbers for the latter. An alternative may be to focus only on the methods and not argue at all about the quality of the SRB data.

In the revised manuscript I have focused only on the methods and do not argue at all about the quality of the SRB data.

Figure 1: Which of the differences are statistically significant?

This figure has been removed from the manuscript, in line with focusing on the methods.

Table 1: How about the altitude dependence of short wave radiation? (See e.g. Marty, Philipona, Frohlich, Ohmura, Theor. Appl. Climatol. 2002)

Also this table has been removed from the manuscript (and along with it the question of how shortwave radiation changes with altitude), in line with focusing on the methods.

p.6, l.6: What do you mean by bilinear interpolation from coarse (SRB) to fine (E2OBS) grid? Copying? Same question on p.11, l.18.

I have changed "bilinearly interpolated" to "spatially bilinearly interpolated" in both cases. I think this is a standard term, which does not need further explanation.

p.6, l.8: "For the BCvtp2 methods, the sub-SRB-grid scale spatial structure of the original E2OBS data is imposed upon spatially disaggregated SRB data prior to bias correction at the E2OBS grid." Please try to clarify. I think I understood much later, in Section 3.2.1, that you adjust the mean and variance of E2OBS data on the E2OBS grid with mean and variance of SRB data on the corresponding, coarser SRB gird. True?

I have changed this sentence to "the BCvtp2 methods adjust mean values and variances at the E2OBS grid such that mean values and variances of spatial aggregates to the SRB grid match the corresponding SRB estimates while the sub-SRB-grid scale spatial structure of mean values and variances present in the original E2OBS data is retained."

p.6, l.14: "... of the underlying four E2OBS values." The two grids thus are such that four E2OBS cells correspond to one SRB cell? They are not shifted against each other?

Correct. I have added the sentence " Every SRB grid cell contains exactly four E2OBS grid cells." to the data description section.

p.6, l.16: It would be helpful if you added some information, possibly equations, on transfer functions, target distributions, estimation of means and variances of beta functions etc. in an appendix, as these are absolutely central to your study. Currently, the reader has to know all this or has to check out the references. After all, you even devote an appendix to explaining Kolmogorov-Smirnov.

Thank you for pointing this out. Such an appendix has been added to the revised manuscript.

Figure 2d: Why are the colored lines so far away from the black and gray lines?

Because my estimates of the upper bounds of monthly mean radiation are calculated based on the upper bounds to the corresponding daily mean radiation. The resulting upper bounds are typically

much larger than observed maximum monthly mean radiation because 31 consecutive days of daily mean radiation at its physical upper limit are very unlikely to occur in reality. I have added this explanation to Sect. 3.1.2.

p.7, l.8: What do you mean by "The rsdt climatology at a given latitude is rescaled such that it sits just above the multi-year maximum..."? Why do that?

To answer your questions, I have rewritten the beginning of this paragraph as follows: "The BCsda1 method employs the climatology of daily mean shortwave insolation at the top of the atmosphere (rsdt; see Appendix B for how rsdt is calculated in this study) for the upper bound estimation. This is motivated by rsds being limited by rsdt in most locations and seasons, which suggests that the annual cycle of the upper bound of daily mean rsds has a similar shape as the climatology of daily mean rsdt. Therefore, method BCsda1 uses a rescaled daily mean rsdt climatology as the upper bound climatology of daily mean rsds (solid blue line in Fig. 1c). The rescaling is done with the smallest possible factor which guarantees that the resulting upper bounds are greater than or equal to the multi-year maximum values of daily mean rsds on all days of the year with rsdt  $\geq$  50 W m-2. An extension of this guarantee to days of the year with lower rsdt would inflate the rescaling factor because during dusk and dawn of polar night, rsds can exceed rsdt due to diffuse radiation coming in from lower latitudes. Therefore, on days of the year with rsdt  $\leq$  50 W m-2, the maximum of the rescaled rsdt and the empirical multi-year maximum daily mean rsds is used as the upper rsds bound."

p.10, l.9: "... one possibility to define ..." What would other possibilities be? Why your choice?

Another possibility would be to follow the BCvtp0 approach, i.e. to use interpolated data. The motivation of my choice is that it solves the problem illustrated and discussed in Sect. 4.2. I have rephrased the sentence as follows: "With target distributions fixed at the SRB grid, target distributions at the E2OBS grid can be defined such that the bias-corrected data have the SRB-grid scale target distributions and the sub-SRB-grid scale structure of the original E2OBS data."

p.10, Eq. 1: Where does the equation come from? Can you give a reference? The explanation following eq. 1 reads rather lengthy but not too clearly.

This does not need any reference. It is the standard formula for the variance of a linear combination of random variables. I have however inserted one intermediate step using covariances in the equation to make its derivation easier to understand.

p.11, l.9: How often does this "99%" condition kick in?

For longwave (shortwave) radiation, this "99%" condition kicks in over four (11% of all) grid cells and there on 15% (5%) of all days of the year. I have added this information to the revised manuscript version.

p.11, l.16: How often does this "40%" condition kick in?

The "40%" condition is never met for longwave radiation whereas for shortwave radiation it kicks in over 14% of all E2OBS grid cells and there on 2% of all days of the year. I have added this information to the revised manuscript version.

p.11, l.27: "Metrics used..." Why these? Why, for example, skewness? What do I learn from this measure? And why a Kolmogorov-Smirnov test? Why not a test that gives more weight to tails, e.g. Anderson-Darling? More generally, when do you say that your bias adjustment is good? When the

adjusted E2OBS distribution is identical (mean, variance, skewness...) to the SRB distribution? Why then adjust at all and not just take the SRB data? Can you use your method to adjust E2OBS data beyond the time span where SRB data is available?

The skewness is included because it is the first distribution moment which is not explicitly adjusted by my parametric QM methods. It is included here to illustrate this conceptual imperfection of my methods. I have included this motivation in the revised results section. You are right about the KS test and the relatively low weight it gives to tails. In the revised section 4.1, I have included Kuiper's test as one that (like the suggested AD test) gives the same weight to CDF differences at all quantiles. Qualitatively, however, the Kuiper's test results are the same as those of the KS test. You are right that I (and, as far as I know, everybody else who cross-validates bias correction methods) consider a bias adjustment good if the adjusted distributions are identical to the target distributions. I have included this definition of (overall) performance in the revised section 4.1. In the ISIMIP framework, there are two reasons for doing the bias adjustment of E2OBS to SRB data and not just using the SRB data directly: It (i) promises a higher inter-variable consistency (e.g. consistency of temperature and longwave radiation) in the EWEMBI dataset and (ii) produces radiation data that cover a longer time span. Applying the methods to E2OBS data beyond the time span where SRB data are available is fine since the 1979-2013 period is in fact not much larger than the 1983-2007 period, so that the former is expected to be sufficiently well represented by the latter.

p.12, l.2: Does the remark about CVCC imply that your method cannot be used to correct E2OBS data outside the SRB period (1983-2007)?

No, it does not, see my response to your previous comment.

p.12, l.11: "In the following, cross-validation results are only shown and discussed for the BCvtp0 and BCvtp1 methods, since results for the corresponding BCvtp1 and BCvtp2 are virtually identical." What do you mean? That the difference between BCvtp0 and BCvtp1 is similar as between BCvtp1 and BCvtp2? And, consequently, BCvtp0 and BCvtp2 differ more?

No, I mean that cross-validation results for the BCvtp1 and BCvtp2 methods are virtually identical. In order to make this clearer I have rewritten the statement as follows: "Please note that results for BCvtp2 are not shown or discussed in this section because BCvtp1 and BCvtp2 produce virtually identical data at the SRB-grid scale."

p.12, l.17: "... overall performance ..." What do you mean by overall performance?

This is now better explained in the new section 4.1.3.

p.12, l.24: Why now looking at relative differences?

Why not? For standard deviations, I think this makes more sense than to look at absolute differences.

Figure 3: I guess a good bias correction in your metrics results in a white map. True? The color / hue coding may be better explained upon first use.

Not true. White means low agreement in bias direction (positive or negative bias) over months and validation data samples. In order to make this clearer I have added the following sentence to the caption of this figure: "More saturated colours indicate higher statistical significance of biases remaining after bias correction."

Figures 4 and 5: Why are the quantities shown of interest? And, again, what is good and what is bad? If white means "good", then none of the methods performs well here?

The cross-validation of multi-year maximum values shall reveal if it is worthwhile and if so, then how to explicitly adjust upper radiation bounds. This is now better explained in section 4.1.2. For why skewness is of interest, see my response to your comment on p.11, l.27. As to the significance of Figure 5, see my answer to your next question. In terms of what white means, see my answer to your previous comment. The methods are clearly not perfect but I also did not expect that. It does not make sense to make an absolute statement such as "this shows that the method performs well." The only sensible question is if one method performs better than another one. Figures 2 to 4 (formerly 3 and 5) quantify the magnitude of biases of selected statistics that remain after bias correction with different methods.

p.14, l.15: Why should bias adjustment on monthly timescales outperform daily bias adjustment with subsequent monthly averaging?

Because of what is shown in Figure 4 (formerly 5). I have revised the explanation of Figure 4 earlier in the text as follows in order to answer your question: "By design, the BCvdpx and BCvmpx methods are equally good at correcting multi-year mean values of daily mean radiation. However, both day-to-day and year-to-year variability are expected to be differently well corrected by the methods operating at different time scales. Since day-to-day variability is (not) explicitly adjusted by the methods operating at the daily (monthly) time scale the BCvdpx methods are expected to perform better at the daily time scale than the BCvmpx methods. The year-to-year variability, on the other hand, is explicitly corrected by the BCvmpx methods and it is not by the Bcvdpx methods because daily data from different years are pooled before quantile mapping is carried out at the daily time scale. Consequently, biases in interannual standard deviations of monthly mean radiation are much larger after bias correction with BCvda1 than with BCvma1 (Fig. 4), and the BCvmpx methods."

p.15, l.3: "Rather, the p-value distributions depicted in Fig. 6b,d suggest that if sampling errors are taken into account then the BCvdp1 methods correct the distributions of monthly mean values almost as well as the BCvmp1 methods." I do not see this point from the text and / or figure.

**I have removed this sentence from the revised manuscript.**

p.15, l.7: "For BCvdp1, this is linked to an insufficient adjustment of third-and higher-order moments..." Not sure what you mean. That you should use another parametric method that takes into account higher moments? At what point do you start to "overfit" if you do this?

I mean that my parametric methods explicitly adjust mean values and variances. Higher-order moments are only implicitly (and therefore most likely not perfectly) adjusted through the distribution fitting. In fact, with my methods you cannot overfit in your sense because both the normal and the beta (provided its bounds have been fixed) distribution only have two parameters, which are fixed once two moments have been fixed. Therefore, they cannot adjust more than two moments explicitly. An alternative would be to use non-parametric QM methods. I think that all of this is now clearer thanks to the appendix about QM and downscaling that has been appended to the revised manuscript.

p.15, l.11: "... correct the upper tail of the rlds and rsds distributions." Can you say this if you use Kolmogorov-Smirnov, which focuses on the center of the distribution?

Kuiper's test results confirm this. I have adjusted the statement accordingly.

Section 4.2: Comparison with BSRN data. Here you compare point data with area mean data. This comes with potentially quite some uncertainty. See e.g. papers by M.Z. Hakuba et al. 2013 / 2014 / 2016 or N.A.J. Schutgens et al. 2016. Part of your disagreement could have its roots there. More generally, you are looking here more into how good your SRB data is than how good your bias adjustment is. If this is of interest, you should also consider other data, e.g. CERES or global mean estimates for rlds and rsds, e.g. by Trenberth et al. In its current form, the comparison with BSRN data is rather confusing than helping, I think.

I agree (see above). I have removed this section from the manuscript.

Figures 7 and 8: What is the colored rectangle to the lower left in each panel?

These figures have been removed from the revised manuscript.

p.18, l.1: "... and differences between standard deviation biases generated by BCdsdp0, BCsdp1 and BCsdp2 are in line with cross-validation results." What do you mean?

Irrelevant now that this part has been removed from the revised manuscript.

p.18, l.5: "... which again suggests that biases relative to BSRN after bias correction using SRB data depend more on the corresponding SRB data biases than on the method used for the bias correction." So the BSRN comparison does not make sense?

Irrelevant now that this part has been removed from the revised manuscript.

p.18, l.8: I do not understand this paragraph.

Irrelevant now that this part has been removed from the revised manuscript.

p.19, l.1 to 14: I think much of what you are describing here has to do with the fact that you are comparing point measurements with area means. See the above mentioned papers by Hakuba, Schutgens, and references therein.

Maybe. Has been removed from the revised manuscript.

p.19, l.26: Why use a staggered grid?

Smaller differences between RMSDs of adjusted E2OBS data from SRB-grid cell and staggered SRB-grid cell mean values are considered to indicate a better bridging of the E2OBS-to-SRB spatial scale gap. Ideally, there would be no such difference and it would therefore be impossible to tell from this analysis if the target distributions of the bias correction were defined on the SRB or staggered SRB grid. I have included this explanation in the revised manuscript.

Figure 10: The figure seems to suggest that variability is strongly enhanced (red areas) by the bias adjustment. True?

**True.**

Appendix C: What is the take home message? Figure C2 seems to suggest that the window length is irrelevant. True?

True.

**List of all relevant changes made in the manuscript**

- the old Table 1 and the old Figures 1, 7, 8 and 9 have been removed
- all text related to the validation against BSRN data has been removed
- Figures 1 (formerly 2) and 5 (formerly 6) have been simplified
- cross-validation using Kuiper's two-sample test has been added (new Figure 6)
- the results and conclusions sections have been almost entirely rewritten
- a new Appendix A on quantile mapping and statistical downscaling has been added
- the individual reasons for testing the different bias correction methods are better explained
- a more mathematical notation has been introduced to define the bias correction methods in Table 1 (formerly 2)

[revised manuscript text omitted]

30 and for rlds and release 3.0, respectively for rsds. These data cover the whole globe on a regular  $1.0^{\circ} \times 1.0^{\circ}$  latitude-longitude grid and span the 07/1983–12/2007 time period. For bias correction and cross-validation, a 24-year subsample of these data is

used which was used and is used here that spans the 12/1983–11/2007 time period. Additional data from the adjacent months 11/1983 and 12/2007 are employed for computations of running mean values. The SRB estimates of rlds and rsds are based on satellite-derived cloud parameters and ozone fields, reanalysis meteorology and a few other ancillary datasets. Due to a lack of satellite coverage during most of the 07/1983–06/1998 time period over an area centred at 70°E, SRB data artefacts are present

5 over the Indian Ocean (https://gewex-srb.larc.nasa.gov/common/php/SRB\_known\_issues.php; cf. Fig. 1). Deviations of E2OBS from SRB (left) and SRB from SRBQC (right) 12/1983–11/2007 mean longwave (top) and shortwave (bottom) radiation. Root-mean-square deviations (RMSDs) over all ocean and all land grid cells are given at the bottom of each panel.

Deviations of Figs. 2-4, 7). Every SRB grid cell contains exactly four E2OBS from SRB long-term mean rlds and rsds are

- 10 shown in Fig. 1, together with corresponding deviations of SRB from SRB Release 3.0 quality-check (SRBQC)products. The SRB and SRBQC products were produced with different algorithms (Stackhouse Jr. et al., 2011). Since the primary-algorithm products are more reliable than the quality-check products (Zhang et al., 2015; Stackhouse Jr. et al., 2011) the former were used for the bias correction of E2OBS rlds and rsds for EWEMBI. Over land, differences in long-term mean radiation between E2OBS and SRB are greater in magnitude than those between SRB and SRBQC. Over the ocean, the differences are of similar
- 15 magnitude. If deviations of SRB from SRBQC data quantify methodological uncertainty inherent to the SRB data then these findings justify the bias correction of E2OBS rlds and rsds using SRB data over land at least. grid cells.

**2.3 **BSRN**

Observations made at the following 54 BSRN stations are used in this study. In order to adjust rlds for elevation differences between BSRN stations and E2OBS-grid cells, prior to data comparison, BSRN rlds values are offset by the values listed in

- 20 the rightmost column, based on the formula proposed by Stackhouse Jr. et al. (2011; see text). station latitude longitude offset ALE 82.451 - 62.508 - 6.580 ASP - 23.798 133.888 - 4.256 BAR 71.323 - 156.607 0.000 BER 32.267 - 64.667 - 0.112 BIL 36.605
  -97.515 - 0.560 BON 40.060 - 88.370 - 0.280 BOU 40.048 - 105.007 - 9.772 BRB - 15.601 - 47.713 - 0.504 CAB 51.971 4.927
  -0.056 CAM 50.217 - 5.317 0.588 CAR 44.083 5.059 - 14.840 CLH 36.905 - 75.713 0.896 CNR 42.816 - 1.601 - 3.500 COC
  -12.193 96.835 0.140 DAA - 30.665 23.993 0.476 DAR - 12.425 130.891 0.812 DOM - 75.100 123.383 0.534 DRA 36.626
  25 -116.018 - 3.780 EUR 79.980 - 85.930 - 5.740 FLO - 27.533 - 48.517 - 9.324 FPE 48.310 - 105.100 - 1.204 FUA 33.582 130.375
  -1.092 GCR 34.255 - 89.873 - 0.112 GOB - 23.561 15.041 - 4.788 GVN - 70.650 - 8.250 - 0.097 ILO 8.533 4.566 2.492 ISH 24.337
  124.163 - 0.504 station latitude longitude offset IZA 28.500 - 16.300 57.316 KWA 8.720 167.731 0.252 LAU - 45.045 169.689
  -7.420 LER 60.140 - 1.185 1.232 LIN 52.210 14.122 1.764 MAN - 2.058 147.425 - 0.952 MNM 24.288 153.983 0.000 NAU
  -0.521 166.916 -0.084 NYA 78.925 11.950 - 3.388 PAL 48.713 2.208 1.932 PAY 46.815 6.944 - 6.076 PSU 40.720 - 77.930
  30 0.028 PTR - 9.069 - 40.320 0.504 REG 50.205 - 104.713 0.336 SAP 43.060 141.328 - 4.200 SBO 30.860 34.779 - 1.764 SMS
  -29.443 - 53.823 1.932 SON 47.054 12.958 39.424 SOV 24.910 46.410 - 3.640 SPO - 89.983 - 24.799 0.290 SXF 43.730 - 96.620
  - -0.056 SYO -69.005 39.589 -14.012 TAM 22.790 5.529 -1.120 TAT 36.058 140.126 -0.924 TIK 71.586 128.919 -1.484 TOR 58.254 26.462 -0.028 XIA 39.754 116.962 0.280

Ground observations of longwave downward and shortwave downward (global) radiation made at 54 stations of the Baseline Surface Radiation Network (BSRN; Table 1; König-Langlo et al., 2013) are used as independent validation data for rlds and rsds, respectively. BSRN measurements began at a few stations in 1992. The latest measurements included here are from 2014. Daily mean values of BSRN measurements, which are taken every minute or every few minutes, depending on the

- 5 station, are computed in two steps. First, gaps no longer than 467/11 minutes in the original rlds/rsds time series are filled by linearly interpolation between values right before the beginning and after the end of a gap, as suggested by Schild (2016; for statistics of BSRN data gaps see Roesch et al., 2011). Daily mean values are then calculated for days that are fully covered by these gap-filled values. Prior to data comparison, the resulting BSRN data availability masks are applied to the original and bias-corrected E2OBS time series from the respective E2OBS-grid cells. Additionally, BSRN rlds values are adjusted for
- 10 elevation differences between BSRN stations and E2OBS-grid cells as proposed by Stackhouse Jr. et al. (2011). For elevations  $z_{BSRN}$  of BSRN stations and  $z_{E2OBS}$  of E2OBS-grid cells, BSRN rlds values are offset by  $0.028 (z_{BSRN} z_{E2OBS})$  (cf. Table 1).

**3 Methods**

For the reader who is is not familiar with the concepts of quantile mapping and/or statistical downscaling, a short introduction

- 15 including definitions of relevant terms is given in Appendix A. The parametric quantile mapping methods introduced in the following are named according to the scheme BCvtpx, where v, t, p are used to distinguish between methods for longwave and shortwave radiation (v = l, s) operating at the daily and monthly time scale (t = d, m) using basic and advanced distribution types or parameter estimation techniques (p = b, a). Index x = 0, 1, 2 is used for variants of these methods that differ in how they handle the spatial resolution gap between the SRB and E2OBS . The grids. For the BCvtp0 methods correct E2OBS data
- 20 directly at, the SRB data are spatially bilinearly interpolated to the E2OBS grid using bilinearly and the E2OBS data are then bias-corrected using these interpolated SRB data; this is to mimic the Ruane et al. (2015) approach. For bias correction with the BCvtp1 methods, E2OBS data are spatially aggregated to the SRB grid, the aggregated data are then bias-corrected and the resulting data disaggregated back to the E2OBS grid. For; this approach was used to produce the EWEMBI radiation data. Lastly, the BCvtp2 methods , the adjust mean values and variances at the E2OBS grid such that mean values and variances
- 25 of spatial aggregates to the SRB grid match the corresponding SRB estimates while the sub-SRB-grid scale spatial structure of mean values and variances present in the original E2OBS data is imposed upon spatially disaggregated SRB data prior to bias correction at the E2OBS grid. The bias correction of E2OBS rlds and rsds for EWEMBI was done with methods BClda1 and BCsda1, respectively. retained; this is to overcome the variability deflation induced by the other two approaches. Since the BCvtp0 and BCvtp2 methods are based on the BCvtp1 methods, the latter are introduced first. Readers who are merely
- 30 interested in how the EWEMBI radiation data were produced are informed that methods BClda1 and BCsda1 were used for that purpose.

**Figure 1.** Parameters Estimation of elimatological distributions-parameters of quantile mapping methods used for the bias correction of longwave (**top**) and shortwave (**bottom**) radiation at the daily (**left**) and monthly (**right**) time scale. This example is based on SRB daily mean rlds and rsds data from 79.5° N, 12.5° E and the 12/1983–11/2007 time period. Climatological distribution parameters are estimated based on empirical 24-year mean values (dark grey), standard deviations (light grey range around mean values) and minimum and maximum values (black) of daily mean (**left**) and 31-day running mean (**right**) radiation computed individually for every day of the year. The distribution parameters estimated for the basic (red) and advanced (blue) bias correction methods (cf. Table 1) include mean values and standard deviations (dotted red, dashed blue), and upper bounds (solid red, solid blue) where beta distributions are used. Note that the basic and advanced estimates of mean values and standard deviations only differ in panel (c) near the beginning and end of polar night (cf. Table 1). The light-blue green line in panel (a) represents 25-day running mean values of 25-day running maximum values of 24-year maximum values of daily mean rlds, which are used to estimate the upper bounds of the climatological beta distributions are set to zero. The lowermost and uppermost dotted red and dashed blue lines are the medians of sample minimum and maximum values of random samples of length 24 drawn from the estimated elimatological distributions. This plot is based on SRB daily mean rlds and rsds data from 79.5, 12.5 and the 12/1983–11/2007 time period.

**3.1 Bias correction at the SRB gridSRB-grid scale**

For the BCvtp1 methods, daily mean E2OBS rlds and rsds are first aggregated to the SRB grid using a first-order conservative remapping scheme (Jones, 1999). This The conservative remapping ensures that each aggregated value is the grid-cell area-weighted mean of the underlying four E2OBS values. In the following, the The methods of bias correction of these aggregated

5

values are described --in the following. The method used for the subsequent disaggregation to the E2OBS grid is described in Sect. 3.1.3.

The BC*vtp*1 methods use parametric transfer functions of the form  $F_{vtp}^{\text{SRB}-1}(F_{vtp}^{\text{E2OBS}}(\cdot))$ , where  $F_{vtp}^{\text{E2OBS}}$  and  $F_{vtp}^{\text{SRB}}$  are climatological cumulative distribution functions (CDFs) of aggregated E2OBS and SRB data, respectively, estimated at daily temporal resolution for each. The CDFs are estimated individually for every SRB-grid cell individually and day of the year

**Table 1.** Distribution types and parameter estimation methods of bias correction methods BCvtp1 for day d of the year (cf. Fig. 1). Please note that the lower bounds of all climatological beta distributions are set to zero and that 24-year statistics are replaced by 12-year statistics for cross-validation.

| method | distribution type | mean value $\mu_d$                                                                                              | variance $\sigma_{d}^2$                                                             | upper bound $b_{d_{\sim}}$                                                                                                                                                                                                                                                                                                                                                                                                                                                                                                                                                                                                                                                                                                                                                                                                                                                                                                                                                                                                                                                                                                                                                                                                                                                                                                                                                                                                                                                                                                                                                                                                                                                                                                                                                                                                                                                                                                                                                                                                                                                                                                                                                                                                                                                                                                                                                                                                                                                                                                                                                                                                                                                                                                                                                                                                                                                                                                                                                         |
|--------|-------------------|-----------------------------------------------------------------------------------------------------------------|-------------------------------------------------------------------------------------|------------------------------------------------------------------------------------------------------------------------------------------------------------------------------------------------------------------------------------------------------------------------------------------------------------------------------------------------------------------------------------------------------------------------------------------------------------------------------------------------------------------------------------------------------------------------------------------------------------------------------------------------------------------------------------------------------------------------------------------------------------------------------------------------------------------------------------------------------------------------------------------------------------------------------------------------------------------------------------------------------------------------------------------------------------------------------------------------------------------------------------------------------------------------------------------------------------------------------------------------------------------------------------------------------------------------------------------------------------------------------------------------------------------------------------------------------------------------------------------------------------------------------------------------------------------------------------------------------------------------------------------------------------------------------------------------------------------------------------------------------------------------------------------------------------------------------------------------------------------------------------------------------------------------------------------------------------------------------------------------------------------------------------------------------------------------------------------------------------------------------------------------------------------------------------------------------------------------------------------------------------------------------------------------------------------------------------------------------------------------------------------------------------------------------------------------------------------------------------------------------------------------------------------------------------------------------------------------------------------------------------------------------------------------------------------------------------------------------------------------------------------------------------------------------------------------------------------------------------------------------------------------------------------------------------------------------------------------------------|
| BCldb1 | normal            | $\frac{1}{1} \left( \left\langle x_{ij} \right\rangle_{i24} \right)_{j25d}$                                     | $\frac{1}{125}$ m25ys24 4 ({ $x_{ij}$ } i24 ) i25d | _                                                                                                                                                                                                                                                                                                                                                                                                                                                                                                                                                                                                                                                                                                                                                                                                                                                                                                                                                                                                                                                                                                                                                                                                                                                                                                                                                                                                                                                                                                                                                                                                                                                                                                                                                                                                                                                                                                                                                                                                                                                                                                                                                                                                                                                                                                                                                                                                                                                                                                                                                                                                                                                                                                                                                                                                                                                                                                                                                                                  |
| BClda1 | beta              | $\frac{1}{1} \frac{1}{\sqrt{x_{ij}_{i24}}_{j25d}}$                                                              | $\frac{\text{rm}_{25ys24^{4}}}{(\{x_{ij}\}_{i24})_{j25d}}$                          | $\frac{\text{rm}_{25rx}_{25yx}_{24}-\text{rm}_{25ym}_{24}}{A(\langle x_{jj} \rangle_{j24} \rangle_{j25d} + B}$                                                                                                                                                                                                                                                                                                                                                                                                                                                                                                                                                                                                                                                                                                                                                                                                                                                                                                                                                                                                                                                                                                                                                                                                                                                                                                                                                                                                                                                                                                                                                                                                                                                                                                                                                                                                                                                                                                                                                                                                                                                                                                                                                                                                                                                                                                                                                                                                                                                                                                                                                                                                                                                                                                                                                                                                                                                                     |
| BClmb1 | normal            | $\frac{\text{ym}^24\text{rm}^{31^2}}{(\langle x_{ij} \rangle_{j31d})_{i24}}$                                    | $\frac{\text{ys}24\text{rm}31^5}{\text{(x}_{ij})_{j31d}}_{i24}$                     | —                                                                                                                                                                                                                                                                                                                                                                                                                                                                                                                                                                                                                                                                                                                                                                                                                                                                                                                                                                                                                                                                                                                                                                                                                                                                                                                                                                                                                                                                                                                                                                                                                                                                                                                                                                                                                                                                                                                                                                                                                                                                                                                                                                                                                                                                                                                                                                                                                                                                                                                                                                                                                                                                                                                                                                                                                                                                                                                                                                                  |
| BClma1 | beta              | $\frac{\text{ym}^24\text{rm}^{31^2}}{(\langle x_{ij} \rangle_{j31d})_{i24}}$                                    | $\frac{\text{ys}24\text{rm}31^5}{\text{(x_{ij})_{j31d}}_{i24}}$                     | $\frac{1}{100} \frac{1}{100} \frac{1}$ |
| BCsdb1 | beta              | $\frac{\text{rm25ym24}^{1}}{(\langle x_{ij} \rangle_{i24} \rangle_{j25d}}$                                      | $\frac{\text{rm}_{25ys24^{4}}}{(\{x_{ij}\}_{i24})_{j25d}}$                          | $\frac{1}{10000000000000000000000000000000000$                                                                                                                                                                                                                                                                                                                                                                                                                                                                                                                                                                                                                                                                                                                                                                                                                                                                                                                                                                                                                                                                                                                                                                                                                                                                                                                                                                                                                                                                                                                                                                                                                                                                                                                                                                                                                                                                                                                                                                                                                                                                                                                                                                                                                                                                                                                                                                                                                                                                                                                                                                                                                                                                                                                                                                                                                                                                                                                                     |
| BCsda1 | beta              | $\frac{\text{rm25}^{*}\text{ym24}^{3}}{(\langle x_{ij} \rangle_{i24} \rangle_{j25d^{*}}}$                       | $\frac{rm25^*ys24^6}{\langle\{x_{ij}\}_{i24}\rangle_{j25d^*}}$                      | <del>yx24-rsdt10 Crsdtd</del>                                                                                                                                                                                                                                                                                                                                                                                                                                                                                                                                                                                                                                                                                                                                                                                                                                                                                                                                                                                                                                                                                                                                                                                                                                                                                                                                                                                                                                                                                                                                                                                                                                                                                                                                                                                                                                                                                                                                                                                                                                                                                                                                                                                                                                                                                                                                                                                                                                                                                                                                                                                                                                                                                                                                                                                                                                                                                                                                           |
| BCsmb1 | beta              | $\frac{1}{2} \frac{(\langle x_{ij} \rangle_{j31d})_{i24}}{\langle \langle x_{ij} \rangle_{j31d} \rangle_{i24}}$ | $\frac{\text{ys}24\text{rm}31^5}{\text{(}x_{ij})_{j31d}}_{i24}$                     | $\frac{\text{rm31sdb}^{11}}{\text{(}b_{j}^{\text{sdb1}}\text{)}_{j31d}}$                                                                                                                                                                                                                                                                                                                                                                                                                                                                                                                                                                                                                                                                                                                                                                                                                                                                                                                                                                                                                                                                                                                                                                                                                                                                                                                                                                                                                                                                                                                                                                                                                                                                                                                                                                                                                                                                                                                                                                                                                                                                                                                                                                                                                                                                                                                                                                                                                                                                                                                                                                                                                                                                                                                                                                                                                                                                                                           |
| BCsma1 | beta              | $\frac{1}{2}$ $\frac{\sqrt{x_{ij}}_{j31d}_{i24}}{\sqrt{x_{ij}}_{j31d}_{i24}}$                                   | $\frac{\text{ys24rm31}^5}{\text{(x_{ij})_{j31d}}_{i24}}$                            | $\frac{1}{1} \frac{b_{j}^{\text{sda1}}}{2} \frac{b_{j}^{s$                                                                               |

1 25-day running mean value of 24-year daily mean values

2 24-year-daily mean value of 31-day running mean values, with February 29 value replaced by average of February 28 and March 1 values

3-25-or-fewer-day running mean value of 24-year daily mean values (see text)

4 25-day running mean value of 24-year daily variances

5 24-year daily variance of 31-day running mean values, with February 29 value replaced by average of February 28 and March 1 values

6-25-or-fewer-day running mean value of 24-year daily variances (see text)

7 affine transformation of mean value elimatology of BClda1 that sits just above the 25-day running mean values of 25-day running maximum values of 24-year maximum values of daily mean rlds (see text)

8 31-day running mean value of upper bounds of BClda1 method

9 25-day running mean value of 25-day running maximum values of 24-year maximum values of daily mean rsds

10 resealed rsdt elimatology that sits just above 24-year maximum values of daily mean rsds (see text)

11 31-day running mean value of upper bounds of BCsdb1 method

 $\frac{12}{31}$ -day running mean value of upper bounds of BCsda1 method  $x_{ij}$  is the daily mean rids (for BCltp1) or rods (for BCstp1) on day j of year i.

Brackets  $\langle \cdot \rangle$ ,  $\{\cdot\}$ , and  $[\cdot]$  denote the calculation of sample mean values, variances, and maximum values, respectively.

Bracket subscripts i24, j25d, and  $j25d^*$  indicate that these sample statistics are calculated over years  $i \in \{1, \dots, 24\}$ , over days  $j \in \{d - 12, \dots, d + 12\}$ , and over days  $j \in \{d - 12, \dots, d + 12\}$ , and over days  $j \in \{1, \dots, 24\}$ .

 $j \in \{d-n, \dots, d+n\} \text{ with } n = \min\{12, \max\{n \ge 0 \colon \forall j \in \{d-n, \dots, d+n\} \colon \operatorname{rsdt}_j > 0\}\}, \text{respectively}.$

Constants A, B, and C are determined by  $\arg\min_{A,B'} \sum_{l=1}^{365} (\langle [[x_{ij}]_{i24}]_{i25k} \rangle_{k25l} - A \langle \langle x_{ij} \rangle_{i24} \rangle_{i25l} + B' \rangle_{2}^{2}$

 $\min\{B \ge 0: \forall l \in \{1, \dots, 365\}: A((x_i)) \ge 1, l \ge 1,$

[revised manuscript text omitted]

**4 Results**

The In the following, the bias correction methods introduced above are assessed in a threefold way. First, original and bias-corrected E2OBS data are compared to SRB data cross-validated at the SRB-grid scale using a cross-validation approach.Secondly, they are compared to independent ground observations made at 54 BSRN stations. Thirdly, (Sect. 4.1), and their disaggregation

5 performance is assessed by comparing sub-SRB-grid scale spatial variability before and after bias correction are compared in order to measure the disaggregation performance of all methods.

Data comparisons are done at the daily and monthly time scale in order to identify strengths and weaknesses of bias correction methods operating at either of these time scales. Metrics used to quantify statistical dissimilarity between E2OBS and SRB or BSRN data include differences between multi-year mean values, standard deviations, skewness and maximum

10 values, root-mean-square deviations (RMSDs) between time series, and *p*-values of two-sample Kolmogorov-Smirnov (KS) test statistics for empirical CDF comparisons (see (Sect. C for details4.2).

**4.1 Cross-validation at the SRB-grid scale**

For the cross-validation against SRB data, 24 years worth of overlapping E2OBS and SRB data are divided into two 12-year samples of which the first one is used to calibrate and the second one to validate the method. Switanek et al. (2017) have

- 15 shown that if climatological distributions differ substantially between calibration and validation samples of either the observed (here SRB) or modelled (here E2OBS) data (such differences are hereafter denoted as calibration-validation Common practice would be to use data from the first and second half of the 24-year period to define these samples. Yet due to climate change this definition may yield calibration and validation data samples that differ statistically. These differences in turn, which are essentially climate change signalsor CVCCSs), then the remaining biases after quantile mapping trained on the calibration data
- 20 sample and applied to the validation datasample are dominated by differences between observed and modelled CVCCSs. This implies that calibration and validation data samples should be made as statistically similar as possible if the, may differ in extent between the E2OBS and SRB data. Switanek et al. (2017) have shown that such differences in climate change signals may then dominate cross-validation is to only measure the metrics and thereby distort the comparative validation of bias correction methods' imperfections. Hence. In order to minimise this climate change impact on cross-validation results, here,
- 25 calibration and validation data samples are composed of data from every second and every other year or vice versa, respectively. The samples are accordingly labelled every1st and every2nd.

**4.2 Cross-validation against SRB data**

In the following, cross-validation results are only shown and discussed for the BCvtp0 and Please note that results for BCvtp2are not shown or discussed in this section because BCvtp1 methods since results for the corresponding BCvtp1 and BCvtp2

30 are virtually identical . In order to (i) measure how the use of spatially interpolated SRB data for bias correction impacts produce virtually identical data at the SRB-grid scale biases, and (ii) assess the value of scale.

**4.1.1 BCvtp0 versus BCvtp1**

The first question addressed here is how the bilinear spatial interpolation of SRB data to the E2OBS grid before bias correction with the BCvtp0 methods impacts the extra complications involved in the parameter estimations of the advanced compared to the basic bias correction methods distribution of bias-corrected rlds and rsds values at the SRB-grid scale. To quantify

- 5 these impacts, biases in multi-year daily mean values, standard deviations, skewness and maximum values remaining after bias correction with methods BCvda0, BCvda1 and BCvdb1 are compared first. Then, bias correction methods operating at different temporal scales are compared with respect to their ability to adjust the interannual variability of monthly mean values . Lastly, the overall performance of all BCvtp1 methods is assessed via CDF comparisons at both the daily and monthly time seale. BCvda1 are compared in the left and middle columns of Figs. 2 and 3.
- 10 Maps of biases in multi-year mean values, standard deviations, skewness Since linear interpolation always yields values that are intermediate to the values at the interpolation knots it is expected that daily SRB data bilinearly interpolated to the E2OBS grid and then aggregated back up to the SRB grid will be more smooth overall both in space and time than the original SRB data. Manifestations of the increased smoothness in time are the more negative biases of standard deviations (Fig. 2) and maximum values of daily mean rlds and rsds (Fig. 3) remaining after bias correction at the daily time scale are depicted in
- 15 Figs. 2 and 3. Remaining mean value biases for BCvdp1 are small with medians with BCvda0 than with BCvda1. Standard deviations after bias correction with BCvda0 in particular are negatively biased by more than 4% (median over calendar months and × validation data samplesbeing within ±1 at most locations. At low/high latitudes, BCsdb1 leaves smaller/larger mean value biases than BCsda1. In comparison to BCvda1, BCvda0 leaves greater mean value biases in particular over coastal and mountainous regions, where spatial gradients are large.
- 20 Medians of relative standard deviation biases ) in most regions. In mountainous and therefore spatially heterogeneous regions, also multi-year monthly mean radiation is changed significantly by the interpolation, with median biases over calendar months  $\times$  validation data samples remaining after bias correction with BCvdp1 are mostly within  $\pm 4\%$ . Underestimations by more then 4% remain over large parts of subtropical Northern Hemisphere land. In most locations, BCldb1 leaves smaller rlds standard deviation biases than BClda1. Bias correction with BCvda0 yields systematically too low standard deviations in
- 25 most locations, in particular for shortwave radiation. This is a result of the variance deflation the bilinear interpolation inflicts on the SRB data. exceeding  $2 \text{ Wm}^{-2}$  in many such places (Fig. 2).

Large skewness biases with medians frequently exceeding  $\pm 50\%$  remain

**4.1.2 BCvtax versus BCvtbx**

Next is an assessment of how the treatment of the upper bound of the distributions estimated by the BCvdp1 methods impacts the distribution of bias-corrected rlds and rsds values at the SRB-grid scale. To quantify these impacts, biases in multi-year

30 the distribution of bias-corrected rlds and rsds values at the SRB-grid scale. To quantify these impacts, biases in multi-year daily mean values, standard deviations, and maximum values remaining after bias correction with any method. The median skewness of longwave radiation is mostly too low, in particular over the ocean and no matter if CDFs of beta or normal distributions are used in the transfer function. The median skewness of shortwave radiation is too low over most of the tropics